# Policy-labeled Preference Learning: Is Preference Enough for RLHF?

**Taehyun Cho** [* 1]  **Seokhun Ju** [* 1]  **Seungyub Han** [1]  **Dohyeong Kim** [1]  **Kyungjae Lee** [2]  **Jungwoo Lee** [1 3]

## Abstract

To design rewards that align with human goals, Reinforcement Learning from Human Feedback (RLHF) has emerged as a prominent technique for learning reward functions from human preferences and optimizing policies via reinforcement learning algorithms. However, existing RLHF methods often misinterpret trajectories as being generated by an optimal policy, causing inaccurate likelihood estimation and suboptimal learning. Inspired by Direct Preference Optimization framework which directly learns optimal policy without explicit reward, we propose *policy-labeled preference learning* (PPL), to resolve *likelihood mismatch* issues by modeling human preferences with *regret*, which reflects behavior policy information. We also provide a contrastive KL regularization, derived from regret-based principles, to enhance RLHF in sequential decision making. Experiments in high-dimensional continuous control tasks demonstrate PPL's significant improvements in offline RLHF performance and its effectiveness in online settings. For more information, visit our project page: https://jjush.github.io/PPL/.

## 1. Introduction

Preference-based reinforcement learning (PbRL), a branch of RLHF, focuses on learning optimal policies directly from human preferences, avoiding the needs for explicit, handcrafted rewards. Unlike traditional RLHF, which infers numerical rewards, PbRL derives reward signals from preference comparisons between trajectory pairs. As RLHF applications expand into complex domains such as large language models (LLMs) (Achiam et al., 2023; Touvron et al., 2023; Ye et al., 2024; Meng et al., 2024; Xiao et al., 2024) and

robotic manipulation (Christiano et al., 2017; Hwang et al., 2023), aligning agents with human intentions has become increasingly important.

Early RLHF research (Lee et al., 2021; Park et al., 2022) assumed humans prefer trajectories with higher cumulative rewards, leading to a two-step learning process: (1) training a *reward* model to align with human preferences and (2) applying a RL algorithm to optimize the *policy* using the learned reward. Recently, Rafailov et al. (2024b) introduced Direct Preference Optimization (DPO), which bypasses the need for an explicit reward function and directly optimizes the policy based on preferences. This simplification reduces computational complexity and dependency on potentially imperfect reward models, improving training stability. DPO has demonstrated superior performance over reward-based RLHF methods, particularly in fine-tuning LLMs (Bai et al., 2022; Stiennon et al., 2020; Ziegler et al., 2019) and offline RL benchmarks (Hejna & Sadigh, 2024).

While DPO has shown strong performance in LLM fine-tuning and offline RL benchmarks, its assumptions are largely shaped by the structure of LLM training. Many RLHF studies assume contextual bandits or deterministic Markov decision processes (MDPs), where the next state is determined solely by the previous state (prompt) and action (response), leading to a simplified transition model. However, in standard RL settings, state transitions involve *environmental stochasticity*, introducing additional uncertainty that complicates both policy optimization and MDP estimation (Yang et al., 2022). Since outcomes depend on external stochasticity beyond the agent's control, observed transitions may not always accurately provide the agent's policy performance. Consequently, inferring optimal behavior from observed sequences becomes more challenging. For example, an agent executing a suboptimal policy might transition to a preferred state with low probability, whereas an optimal policy could occasionally lead to an undesirable state due to environmental randomness. This contrast with the deterministic nature of LLM training, where responses are directly mapped from prompts, highlights a key limitation when applying DPO to general RL problems.

This challenge becomes more critical in offline RL, where learning relies on pre-collected datasets rather than direct interaction with environment. When data comes from di-

*Equal contribution [1]Seoul National University, Seoul, South Korea [2]Korea University, Seoul, South Korea [3]HodooAI Labs, Seoul, South Korea. Correspondence to: Kyungjae Lee <kyungjae_lee@korea.ac.kr>, Jungwoo Lee <junglee@snu.ac.kr>.

*Proceedings of the 42$^{nd}$ International Conference on Machine Learning*, Vancouver, Canada. PMLR 267, 2025. Copyright 2025 by the author(s).

verse policies but lacks explicit information about the behavior policies, distinguishing whether outcome quality stems from policy suboptimality or environmental stochasticity becomes difficult. Given this ambiguity, a key question arises:

*Can preference data generated by diverse policies sufficiently guide sequential decision-making, or is additional information required?*

To address this question, we propose a novel RLHF framework, *Policy-labeled Preference Learning* (PPL), which leverages regret-based preference modeling while explicitly labeling the behavior policy. Unlike conventional approaches that rely solely on preferences between trajectory pairs, PPL incorporates policy information directly into the learning process, disentangling the effects of environmental stochasticity and the suboptimality of behavior policies.

To provide theoretical insights, we define a *reward equivalence class*—a set of reward functions that induce the same optimal policy—and derive a bijective mapping that allows regret to be expressed as a function of the optimal policy. We show that, unlike the partial sum of rewards, regret is uniquely defined with respect to a given optimal policy, making it a well-structured metric that mitigates issues related to reward sparsity and enhances the stability of learning. Furthermore, we introduce *contrastive KL regularization*, which sequentially aligns policies with preferred trajectories while explicitly contrasting them against less preferred ones. Empirically, to consider the fact that real-world offline data often consists of rollouts from diverse policies, we construct homogeneous and heterogeneous datasets in the MetaWorld environment and evaluate performance across various offline datasets.

## 2. Preliminaries

**Maximum Entropy Framework.** We define the MDP as $\mathcal{M} = (\mathcal{S}, \mathcal{A}, \mathbb{P}, r, \gamma)$ characterized by state space $\mathcal{S}$, action space $\mathcal{A}$, transition kernels $\mathbb{P}$ which represents the probability of the next state $s'$ given the current state $s \in \mathcal{S}$ and action $a \in \mathcal{A}$, underlying reward $r \in [r_{\min}, r_{\max}]$, and discount factor $\gamma$. For notational simplicity, we denote the expectation over trajectories $\tau = (s_0, a_0, s_1, a_1, \cdots)$ generated by a policy $\pi$ and the transition kernel $\mathbb{P}$ as $\mathbb{E}_{\tau \sim \mathbb{P}^\pi}[\cdot]$.

The MaxEnt framework provides an optimal policy which not only maximizes the expected cumulative return, but also the entropy for each visited state:

$$\pi^*_{\text{MaxEnt}} = \arg\max_\pi \mathbb{E}_{\tau \sim \mathbb{P}^\pi}\Big[\sum_{t \geq 0} \gamma^t (r(s_t, a_t) + \alpha \mathcal{H}^\pi(\cdot|s_t))\Big],$$

where $\mathcal{H}^\pi(\cdot|s) = -\mathbb{E}_\pi[\log \pi(\cdot|s)]$ is the entropy of policy $\pi$ at state $s$. Here, $\alpha$ is a temperature hyperparameter that determine the relative importance of entropy and reward.

Table 1: Comparison for different preference models under PbRL framework.

| Algorithm | Score Function | Direct Preference Optimization | Likelihood Matching |
|---|---|---|---|
| **PEBBLE** (Lee et al., 2021) | $r_\psi(s_t, a_t)$ | ✗ | ✗ |
| **DPO** (Rafailov et al., 2024b) | $\log \pi_\psi(y\|s)/\pi_{\text{ref}}(y\|s)$ | ✓ | ✗ |
| **DPPO** (An et al., 2023) | $-\mathbb{E}_{a \sim \pi_\psi(\cdot\|s_t)}[\|\|a - a_t\|\|_2]$ | ✓ | ✗ |
| **CPL** (Hejna et al., 2023) | $Q^{\pi_\psi}(s_t, a_t) - V^{\pi_\psi}(s_t)$ | ✓ | ✗ |
| **PPL** [Ours] | $-(V^{\pi_\psi}(s_t) - Q^{\pi}(s_t, a_t))$ | ✓ | ✓ |

For clarity, we say $\pi^*_{\text{MaxEnt}}$ as $\alpha$-optimal. In addition, soft $Q$-function $Q^\pi(s, a)$ is defined as the expected cumulative return augmented by an entropy terms, expressed as;

$$Q^\pi(s, a) = r(s, a) + \mathbb{E}_{\tau \sim \mathbb{P}^\pi}\Big[\sum_{t > 0} \gamma^t (r(s_t, a_t) + \alpha \mathcal{H}^\pi(\cdot|s_t))\Big].$$

Analogously, we can derive soft value function $V^\pi(s)$ and soft Bellman equation as follows:

$$V^\pi(s) = \mathbb{E}_{a \sim \pi}[Q^\pi(s, a) - \alpha \log \pi(a|s)],$$
$$Q^\pi(s, a) = r(s, a) + \gamma \mathbb{E}_{s' \sim \mathbb{P}}\Big[V^\pi(s')\Big]$$

for all state-action pairs $(s, a) \in \mathcal{S} \times \mathcal{A}$. Note that the interpretation of the value function is modified by involving the entropy term in the MaxEntRL, *i.e.*, $V^\pi(s) \neq \mathbb{E}_\pi[Q^\pi(s, a)]$. For an $\alpha$-optimal policy $\pi^*$, Ziebart (2010) derived the relationship between the optimal policy and optimal soft $Q$-function $Q^{\pi^*}$:

$$\pi^*(a|s) = \exp\Big(\alpha^{-1}(Q^{\pi^*}(s, a) - V^{\pi^*}(s))\Big),$$
$$V^{\pi^*}(s) = \alpha \log \int_{a \in \mathcal{A}} \exp\Big(\alpha^{-1} Q^{\pi^*}(s, a) da\Big).$$

### 2.1. Preference-based Reinforcement Learning

Designing a reward function that accurately aligns with human behaviors is inherently challenging. To address this, PbRL focuses on learning the optimal policy directly from human preferences rather than relying on predefined rewards. In this context, we adopt a reward-free MDP $\mathcal{M} \setminus r$ within the MaxEnt framework. We define a segment $\zeta = (s_0, a_0, \ldots, s_k, a_k)$ as a sequence sampled from a dataset $\mathcal{D}$. Specifically, human annotators or AI systems are tasked with comparing pairs of trajectory segments $(\zeta^+, \zeta^-)$, where $\zeta^+$ is preferred over $\zeta^-$ (*i.e.*, $\zeta^+ \succ \zeta^-$).

**Score-based Preference Model.** Score-based preference model is a natural generalization of RLHF for modeling human preferences through score functions, instead of partial

sum of rewards (Lee et al., 2021). This approach extends the Bradley-Terry model (Bradley & Terry, 1952), where pairwise comparisons are used to infer relative preferences, by introducing a score function that evaluates all observed state-action pairs within a segment. The preference model then assigns probabilities proportional to the sum of these scores, aligning with the Bradley-Terry framework. To implement the preference model using a neural network, the score function is parametrized as $S_\psi$, and the model is trained by minimizing the cross-entropy loss between its predictions and the preference labels derived from the dataset $\mathcal{D}$, as follows:

$$P_{S_\psi}[\zeta^+ \succ \zeta^-] = \sigma\Big(\sum_{t \geq 0} S_\psi(s_t^+, a_t^+) - S_\psi(s_t^-, a_t^-)\Big),$$

$$\mathcal{L}(S_\psi; \mathcal{D}) = -\mathbb{E}_{(\zeta^+, \zeta^-) \sim \mathcal{D}}\Big[\log P_{S_\psi}[\zeta^+ \succ \zeta^-]\Big],$$

where $\sigma(x) = 1/(1 + e^{-x})$ and each $(s_t^+, a_t^+)$ and $(s_t^-, a_t^-)$ is the $t$-th state and action of preferred segment $\zeta^+$ and less preferred segment $\zeta^-$, respectively. For notational simplicity, we abbreviate $\mathbb{E}_{(\zeta^+, \zeta^-) \sim \mathcal{D}}$ as $\mathbb{E}_{\mathcal{D}}$.

Although it is unclear how humans evaluate their preferences, preference models can be improved to better align with human judgment by refining them based on intuitive examples. If the score function does not align with human preference evaluation, the model may produce counterintuitive outcomes. For example, Knox et al. (2022) demonstrated that using the partial sum of rewards as a score function overlooks a critical issue in sparse reward MDPs: all segments that fail to reach the goal are treated as equally preferable, regardless of their contributions.

As shown in Figure 1, sparse reward MDPs provide little feedback for the states that do not reach the terminal goal, leading to meaningless comparison of the preferences in the early- and mid-stage segments based solely on return sums. In contrast, regret is more evenly distributed across timesteps, making it a more effective score for comparing segment preferences regardless of their position in the trajectory. This highlights the importance of modeling preference with the score function that aligns with human intuition. Various approaches to designing such score functions have been proposed, as summarized in Table 1.

**Optimal Advantage-based Preference Model.** Hejna et al. (2023) proposed *Contrastive Preference Learning* (CPL), which is based on an optimal advantage-based preference model (Knox et al., 2022), treating as a regret-based preference model. The CPL score is defined as the difference between the value of the action taken and the average value under the optimal policy, (*i.e.,* $A_{\pi^*}(s_t, a_t) := Q^{\pi^*}(s_t, a_t) - V^{\pi^*}(s_t) = \alpha \log \pi^*(a_t|s_t)$.) Leveraging the relationship between the optimal advantage and the optimal policy within the MaxEnt framework, their objective can

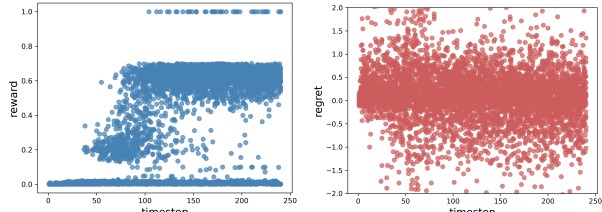

Figure 1: Visualization of 5000 samples in `Bin-Picking-v2` environment. While the ground-truth reward **(left)** is sparse and mainly provided upon task completion, regret **(right)** is more evenly distributed across all timesteps, making it a more informative score function for partial trajectory evaluation.

be reformulated into a policy-based expression, enabling the optimal policy to be learned directly without relying on reward:

$$\mathcal{L}_{\text{CPL}(\lambda)}(\pi_\psi; \mathcal{D}) \tag{1}$$

$$= -\alpha\mathbb{E}_{\mathcal{D}}\Bigg[\log \sigma\Big(\sum_{t \geq 0} \log \pi_\psi(a_t^+|s_t^+) - \lambda \log \pi_\psi(a_t^-|s_t^-)\Big)\Bigg].$$

However, the standard score-based preference loss is convex but not strictly convex, leading to the existence of multiple optimal solutions. Hejna et al. (2023) identified that the shift-invariance property of the loss function (*i.e.,* $P_{S(\pi_\psi)+C} = P_{S(\pi_\psi)}$) causes out-of-distribution actions to be overly weighted, deteriorating learning performance. To mitigate this issue, they introduced an asymmetric regularizer $\lambda$, which reduces the gradient weight on less preferred actions, breaking the inherent symmetry and stabilizing the learning process.

## 3. Policy-labeled Preference Learning

This section introduces the *regret-based preference model* and its distinctions from prior work, with a focus on the issue of *likelihood mismatch*, where sampled segments are misinterpreted as optimal, leading to suboptimal learning. To address this, we propose **Policy-labeled Preference Learning (PPL)**, which employs a regret-based model to accurately estimate segment likelihoods. Finally, we present theoretical results derived from the PPL framework.

### 3.1. Is Preference Enough for RLHF?

**Negative Regret vs Optimal Advantage.** In prior work, Hejna et al. (2023) utilized the *optimal advantage* as the score in CPL to introduce a *regret*-based preference model. While they presented these two concepts as equivalent, they differ significantly in their precise definitions and implications. Optimal advantage refers to the relative benefit of taking a specific action $a$ under the optimal policy $\pi^*$ (*i.e.,* $Q^{\pi^*}(s, a) - V^{\pi^*}(s)$). In contrast, negative regret cap-

## Policy-labeled Preference Learning

$$\sum_\sigma \left( \log \pi^*(a_t|s_t) - \mathbb{E}_{\tau \sim \mathbb{P}^\pi(s_t, a_t)} \left[ \sum_{l>0} \gamma^l D_{KL}\left( \pi(\cdot|s_l)\|\pi^*(\cdot|s_l)\right) \right] \right)$$

Figure 2: Unlike existing DPO algorithms, PPL aligns segment likelihoods by incorporating behavior policies. It reweights gradients based on closeness to the optimal policy, forming a contrastive learning framework.

tures the performance difference between the behavior policy $\pi$ and the optimal policy $\pi^*$ (*i.e.*, $Q^\pi(s,a) - V^{\pi^*}(s)$). The key difference between these concepts lies in whether the *behavior policy* is incorporated into the score.

From a perspective of regret, the optimal advantage disregards the source of the trajectories and evaluates the actions taken solely based on $Q^{\pi^*}$. Consequently, it implicitly treats all trajectories as if they were generated by the optimal policy. This raises an important question: what impact does this assumption – *treating all behavior polices as optimal* – have on the regret-based learning process?

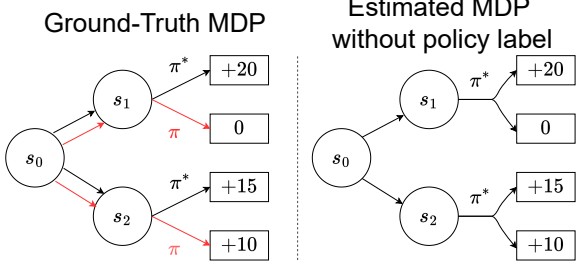

Figure 3: Illustration of the likelihood mismatch problem. Although the behavior policy $\pi$ differs from the optimal policy $\pi^*$, the learning process incorrectly assumes all data is generated by $\pi^*$. As a result, while $\pi^*$ prefers $s_1$, this misinterpretation leads to the incorrect conclusion that $s_2$ is preferred, causing suboptimal learning outcomes.

**Likelihood Mismatch.** Likelihood mismatch occurs when outcome differences between two segments, which actually stem from behavior policy differences, are mistakenly attributed to environmental stochasticity. This misinterpretation leads to incorrect likelihood assignments. Figure 3 illustrates this issue in an offline setting where offline data from both a suboptimal policy $\pi$ and an optimal policy $\pi^*$ lacks explicit policy labels. In this scenario, all data is mistakenly assumed to be generated by the optimal policy $\pi^*$, leading to misinterpretations during learning.

To understand how preference labels are assigned in this setting, let us first consider the left-side figure. The red trajectory, generated by the suboptimal policy $\pi$, assigns

a higher score (+10) to $s_2$, making it appear more preferable than $s_1$. In contrast, the black trajectory, generated by the optimal policy $\pi^*$, assigns a higher score (+20) to $s_1$, leading to the opposite preference. These conflicting results can be properly distinguished when policy labels are available, allowing the model to infer the suboptimality of $\pi$ by evaluating preferences separately for each policy.

Now consider the right-side Figure 3, where the same data is used but without policy labels. Since all data is incorrectly assumed to originate from $\pi^*$, the model observes contradictory outcomes—$s_2$ being preferred in one case and $s_1$ in another—despite assuming a single policy. Lacking policy labels, the model misinterprets this discrepancy as environmental stochasticity rather than differences in policies, distorting the learned MDP and leading to incorrect likelihood estimates for trajectories. To mitigate this issue, it is crucial to explicitly track and incorporate the behavior policy $\pi$ for each segment, ensuring accurate interpretation and proper differentiation of feedback. Thus, replacing optimal advantage with regret, which reflects the suboptimality of the behavior policy, provides a principled solution.

**Regret-based Model Requires the Behavior Policy.** In essence, regret quantifies how much better we could have done if we had followed the optimal policy instead of the behavior policy. A larger regret indicates that the behavior policy is significantly less efficient compared to the optimal policy. We remark that the regret is the difference between the *expected return under optimal policy* and the *achieved return under behavior policy*. Based on the conventional definition of *regret*, we reformulate negative regret in a policy-based form within the MaxEnt framework:

$$
\begin{aligned}
&- \operatorname{Reg}^\pi_{\pi^*}(s_t, a_t) \\
&:= - \underbrace{V^{\pi^*}(s_t)}_{\text{expected return under } \pi^*} + \underbrace{Q^\pi(s_t, a_t)}_{\text{achieved return under } \pi} \quad (2) \\
&\overset{(\text{Thm } 3.4)}{=} \alpha \Big( \underbrace{\log \pi^*(a_t|s_t)}_{\text{increase likelihood}} - \underbrace{\bar{D}_{\text{KL}}\big(\pi\|\pi^*; s_t, a_t\big)}_{\text{decrease sequential forward KL}} \Big).
\end{aligned}
$$

$$(3)$$

In summary, the regret for the preferred segment can be decomposed into two components: First, it increases the likelihood of actions taken in preferred segments, aligning the behavior policy with human preferences. Second, it reduces the sequential forward KL divergence, correcting for likelihood mismatch by considering long-term differences between the behavior policy and the optimal policy. Analogously, for the less preferred segment, the regret exhibits the opposite tendencies. Based on Equation (3), our objective can be formulated as follows:

$$\mathcal{L}_{\text{PPL}}(\pi_\psi; \mathcal{D}) =$$
$$- \mathbb{E}_{\mathcal{D}}\left[\log \sigma\left(-\sum_{t \geq 0} \text{Reg}_{\pi_\psi}^{\pi^+}(s_t^+, a_t^+) - \text{Reg}_{\pi_\psi}^{\pi^-}(s_t^-, a_t^-)\right)\right]$$

where the policy label for the preferred and less preferred segments are denoted as $\pi^+$ and $\pi^-$, respectively. The detailed derivation of this formulation will be introduced in the next section and Appendix B.1.

### 3.2. Theoretical Analysis

Consider a triplet $(\pi^*, (\zeta^+, \pi^+), (\zeta^-, \pi^-))$, where the segments $\zeta^+$ and $\zeta^-$ are generated by policies $\pi^+$ and $\pi^-$, respectively. The (unknown) optimal policy $\pi^*$ serves as the basis for determining the underlying reward and ensuring consistent preferences. During the learning process, we assume that each segment is labeled by its behavior policy. Under this setup, the policy-labeled preference model is expressed as:

$$P_{\pi^*}^{(\pi^+, \pi^-)} = \sigma\Bigg(\sum_{t \geq 0} \underbrace{\left[V^{\pi^*}(s_t^-) - V^{\pi^*}(s_t^+)\right]}_{A_t(\pi^*)}$$
$$+ \underbrace{\left[Q^{\pi^+}(s_t^+, a_t^+) - Q^{\pi^-}(s_t^-, a_t^-)\right]}_{B_t(\pi^*, \pi^+, \pi^-)}\Bigg).$$

This expression is decomposed into two components: (i) $A_t(\pi^*)$, which depends solely on $Q^{\pi^*}$ (note that $V^{\pi^*}(s) = \mathbb{E}_{a \sim \pi^*}[Q^{\pi^*}(s, a)] + \alpha \mathcal{H}^{\pi^*}(\cdot|s)$), and (ii) $B_t(\pi^*, \pi^+, \pi^-)$, which involves $Q^{\pi^+}$ and $Q^{\pi^-}$.

The main theoretical challenge in performing a direct policy update is expressing the soft optimal $Q$-function and soft $Q$-function of a given policy $\pi$ in closed-form with respect to the optimal policy $\pi^*$. Before proceeding, we introduce the concept of *equivalence classes* within the MaxEnt framework to analyze the reward structures that make a given policy optimal.

**Definition 3.1.** The set of reward functions where $\pi^*$ is $\alpha$-optimal is defined as $(\alpha, \pi^*)$-*equivalence class of reward function*, denoted by $\mathcal{R}_{\alpha, \pi^*}$. For every policy $\pi$, the set of $Q^\pi$-function generated by any reward function $r_{\alpha, \pi^*} \in$

$\mathcal{R}_{\alpha, \pi^*}$ is defined as the $(\alpha, \pi^*)$-*equivalence class of $Q^\pi$-function*, denoted by $\mathcal{Q}_{\alpha, \pi^*}^\pi$.

Definition 3.1 indicates that a reward function class $\mathcal{R}$ or a $Q^\pi$-function class $\mathcal{Q}^\pi$ can be partitioned based on the $\alpha$-optimal policy $\pi^*$. For notational simplicity, we denote the ground truth reward function corresponding to the $\alpha$-optimal policy $\pi^*$ as $r_*$ and the $Q^\pi$-function induced by $r_*$ as $Q_*^\pi$, simplifying the subscript to $*$.

**Lemma 3.2** (Structural Condition for $\alpha$-optimality). *A reward function and a soft optimal Q-function where $\pi^*(\cdot|s)$ is $\alpha$-optimal have a one-to-one correspondence with a state-dependent function $\beta : \mathcal{S} \rightarrow \mathbb{R}$, defined as follows:*

$$\mathcal{R}_{\alpha, \pi^*} = \{r_*(s, a) = \alpha \log \pi^*(a|s) + \beta(s) - \gamma \mathbb{E}_{\mathbb{P}}[\beta(s')]\}$$
$$\mathcal{Q}_{\alpha, \pi^*}^{\pi^*} = \{Q_*^{\pi^*}(s, a) = \alpha \log \pi^*(a|s) + \beta(s)\}$$

*for all $s \in \mathcal{S}$ and $a \in \mathcal{A}$.*

Lemma 3.2 demonstrates that the $(\alpha, \pi^*)$-equivalence class of soft optimal $Q$-functions can be uniquely expressed as the sum of a log-probability term, $\alpha \log \pi^*(a|s)$, and a state-dependent function, $\beta(s)$. This result improves upon the prior lemma of Rafailov et al. (2024b), which established only a *surjection* from reward functions to optimal policies. By contrast, we ensure a *bijection*, rigorously defining the equivalence class of reward functions. Furthermore, Lemma 3.2 refines the concept of *policy invariance* introduced by Ng et al. (1999); Gleave et al. (2020) by specifying that the action-dependent term must be $\alpha \log \pi^*(a|s)$ to guarantee $\pi^*$ is the $\alpha$-optimal policy.

**Lemma 3.3** (Unique Fixed Point of Soft Bellman $\pi$-operator). *Let $\pi^*$ be $\alpha$-optimal. For a given policy $\pi$ and $Q$-function $Q_A^\pi \in \mathcal{Q}^\pi$ for any $(s, a) \in \mathcal{S} \times \mathcal{A}$, define the Bellman $\pi$-operator $\mathcal{T}_*^\pi : \mathcal{Q}^\pi \rightarrow \mathcal{Q}^\pi$ where*

$$\mathcal{T}_*^\pi Q_A^\pi(s, a) := Q_*^{\pi^*}(s, a) - \gamma \mathbb{E}_{\mathbb{P}}\Big[\alpha\Big(\mathcal{H}^{\pi^*}(\cdot|s') - \mathcal{H}^\pi(\cdot|s')\Big)$$
$$+ \mathbb{E}_{\pi^*}[Q_*^{\pi^*}(s', a')] - \mathbb{E}_\pi[Q_A^\pi(s', a')]\Big].$$

*Then, $\mathcal{T}_*^\pi$ has a unique fixed point $Q_*^\pi$.*

Lemma 3.3 describes an operator that links the soft $Q$-function of a given policy $\pi$ to the optimal soft $Q$-function $Q_*^{\pi^*}$, identifying $Q_*^\pi$ as its unique fixed point. Notably, this relationship is established without requiring explicit knowledge of the reward function $r_*$. From the novel design of the soft Bellman $\pi$-operator, we now derive the following important theorem.

**Theorem 3.4** (Policy Deviation Theorem). *If a policy $\pi^*$ is $\alpha$-optimal, then for any policy $\pi$,*

$$Q_*^{\pi^*}(s, a) - Q_*^\pi(s, a) = \alpha \bar{D}_{KL}(\pi||\pi^*; s, a)$$

*where the **sequential forward KL divergence** is defined as*

$$\bar{D}_{KL}(\pi||\pi'; s, a) := \mathbb{E}_{\tau \sim \mathbb{P}^\pi_{s,a}}\left[\sum_{l>0} \gamma^l D_{KL}(\pi(\cdot|s_l)||\pi'(\cdot|s_l))\right].$$

*Here, $\mathbb{P}^\pi_{s,a}$ is the distribution of trajectories $\tau = (s_0, a_0, \cdots, s_l, a_l, \cdots)$ generated by policy $\pi$ and the transition $\mathbb{P}$, starting at $(s_0, a_0) = (s, a)$.*

Theorem 3.4 establishes that the difference between the soft $Q$-function of any policy $\pi$ and the optimal soft $Q$-function is constant and can be expressed as the sequential forward KL divergence. Intuitively, $\bar{D}_{KL}(\pi||\pi^*; s, a)$ represents the discounted sum of the forward KL divergence between $\pi$ and $\pi^*$ over the states visited during a rollout starting from $(s, a)$. This property is particularly valuable, as it quantifies the performance gap using only $\pi$ and $\pi^*$.

While related results were proposed by Shaikh et al. (2024) and Zeng et al. (2024), their proofs were restricted to contextual bandits and token-level MDPs with deterministic transitions, respectively. Moreover, their formulation depends on a KL-regularized objective that explicitly incorporates a reference policy. In contrast, Theorem 3.4, formulated within the MaxEnt framework, does not require a reference policy to be well-defined, making it more broadly applicable.

**Corollary 3.5.** *For a given $(\alpha, \pi^*)$ and a policy $\pi$, $Reg^\pi_{\pi^*}(\cdot, \cdot)$ is uniquely determined regardless of $\beta(s)$.*

Since regret is invariant to transformations of $\beta(s)$, it does not require additional variance reduction techniques (Schulman et al., 2015) to ensure stable learning. For a detailed explanation, refer to Appendix B.2.

**Corollary 3.6.** *Maximizing the MaxEnt objective with negative regret as the reward is equivalent to minimizing the sequential forward KL divergence between the learned policy and the behavior policy for each preferred state-action pair in the dataset, i.e.,*

$$\arg\max_{\pi_\psi}\left(\mathbb{E}_{\zeta^+ \sim \mathcal{D}}[-Reg^{\pi^+}_{\pi_\psi}(s^+, a^+) - \alpha \log \pi_\psi(a^+|s^+)]\right)$$

$$\equiv \arg\min_{\pi_\psi}\left(\mathbb{E}_{\zeta^+ \sim \mathcal{D}}[\bar{D}_{KL}(\pi^+||\pi_\psi; s^+, a^+)]\right). \quad (4)$$

Theorem 3.6 implies that regret-based RLHF operates by aggregating behavior policies from preferred segments, aligning the learned policy toward preferred actions. Notably, if all preferred segments are assumed to be generated by the optimal policy, the formulation reduces to the standard CPL objective, highlighting its connection to prior methods. For a more detailed analysis, see Appendix B.3.

### 3.3. Practical Algorithm and Implementation Details

In this section, we present PPL, a practical algorithm that leverages the policy label to solve the likelihood mismatch.

Our setting follows the classical DPO, but with the difference that we manage preference queries by labeling the behavior policy for each trajectory in the dataset. Due to page limitations, see Appendix C for the pseudocode.

**Pseudo-labels.** In general RL settings, the behavior policy that generated a trajectory is typically known or accessible, making policy labeling relatively inexpensive. However, in offline datasets, the behavior policy is often unknown. To address this, we assign *pseudo-labels* as an alternative, assuming each segment was generated by a deterministic policy that executed the observed actions.

**Contrastive KL Regularization.** As previously discussed, the regret is decomposed into two components. In particular, the sequential KL divergence plays a pivotal role in aligning the learned policy with the preferred policy while diverging from the less preferred policy:

$$-\sum_{t\geq 0} \text{Reg}^{\pi^+}_{\pi_\psi}(s^+_t, a^+_t) - \text{Reg}^{\pi^-}_{\pi_\psi}(s^-_t, a^-_t)$$

$$= \alpha \sum_{t\geq 0}\left(\log \frac{\pi_\psi(a^+_t|s^+_t)}{\pi_\psi(a^-_t|s^-_t)}\right.$$

$$\left.\underbrace{- \bar{D}_{KL}(\pi^+||\pi_\psi; s^+_t, a^+_t) + \bar{D}_{KL}(\pi^-||\pi_\psi; s^-_t, a^-_t)}_{\text{contrastive KL regularization } \mathcal{R}(\pi_\psi; \pi^+, \pi^-)}\right).$$

We call this term as *contrastive KL regularization*, which requires performing rollouts for each $(s_t, a_t)$ with respect to $\pi^+$ or $\pi^-$. This regularization term ensures that the learned policy $\pi_\psi$ aligns more closely with the preferred policy $\pi^+$ while pushing away from the less preferred policy $\pi^-$.

In practice, implementing contrastive KL regularization can result in a computational overhead, as it requires multiple rollouts with each state-action pair as the initial point at every timestep until the terminal is reached. This approach can also increase memory usage as it requires additional timesteps outside of the sampled segment. To address these technical challenges, we replace the discounted sum with an $L$-horizon undiscounted sum. We normalize the contrastive KL regularization to balance their scale, and the process is further simplified by reusing segments $\zeta^+, \zeta^-$ as a single rollout of policy $\pi^+, \pi^-$, respectively.

$$\mathcal{R}(\pi_\psi; \pi^+, \pi^-)$$

$$\approx \frac{1}{L}\sum_{l=1}^{L}\left[-\log\frac{\pi^+(a^+_{t+l}|s^+_{t+l})}{\pi_\psi(a^+_{t+l}|s^+_{t+l})} + \log\frac{\pi^-(a^-_{t+l}|s^-_{t+l})}{\pi_\psi(a^-_{t+l}|s^-_{t+l})}\right].$$

Here, $L$ corresponds to the step of look-ahead during rollouts. When $L = 0$, the framework fully reduces to CPL, which does not account rollout for sequential planning. Another interesting observation is that if we assume the segments in the offline dataset were generated by the reference policy (i.e., $\pi^+, \pi^- = \pi_{\text{ref}}$), the framework recovers

Table 2: Success rates of all methods across six tasks on the MetaWorld benchmark on different datasets. Each score is reported with the maximum average performance across four seeds over 200 episode evaluation window.

|  |  | Bin Picking | Button Press | Door Open | Drawer Open | Plate Slide | Sweep Into |
|---|---|---|---|---|---|---|---|
| Homogeneous Dense | SFT | $39.7 \pm 19.2$ | $71.5 \pm 3.3$ | $\mathbf{48.0 \pm 15.6}$ | $56.2 \pm 1.8$ | $64.8 \pm 0.8$ | $70.0 \pm 6.5$ |
|  | P-IQL | $62.0 \pm 4.4$ | $72.3 \pm 1.0$ | $47.7 \pm 5.1$ | $58.0 \pm 5.7$ | $\mathbf{70.5 \pm 6.1}$ | $65.8 \pm 1.3$ |
|  | CPL | $22.7 \pm 5.5$ | $64.3 \pm 1.4$ | $29.0 \pm 4.3$ | $54.0 \pm 4.3$ | $65.5 \pm 3.1$ | $69.8 \pm 3.3$ |
|  | PPL | $\mathbf{83.5 \pm 4.4}$ | $\mathbf{79.8 \pm 4.8}$ | $39.3 \pm 2.0$ | $\mathbf{69.2 \pm 5.5}$ | $64.7 \pm 2.0$ | $\mathbf{72.8 \pm 4.8}$ |
| Homogenous Sparse | SFT | $33.5 \pm 5.4$ | $67.4 \pm 1.5$ | $31.3 \pm 2.1$ | $54.9 \pm 2.7$ | $67.1 \pm 3.7$ | $78.3 \pm 2.5$ |
|  | P-IQL | $72.4 \pm 6.6$ | $74.5 \pm 0.0$ | $\mathbf{58.5 \pm 1.4}$ | $51.4 \pm 4.6$ | $\mathbf{76.3 \pm 1.6}$ | $\mathbf{79.0 \pm 2.6}$ |
|  | CPL | $26.5 \pm 1.0$ | $63.7 \pm 1.3$ | $28.5 \pm 5.8$ | $50.1 \pm 4.5$ | $65.1 \pm 2.8$ | $72.9 \pm 6.1$ |
|  | PPL | $\mathbf{87.2 \pm 3.5}$ | $\mathbf{87.3 \pm 2.8}$ | $49.3 \pm 6.5$ | $\mathbf{68.5 \pm 5.3}$ | $64.0 \pm 6.4$ | $73.9 \pm 3.5$ |
| Heterogeneous Dense | SFT | $18.5 \pm 23.8$ | $63.7 \pm 12.2$ | $26.0 \pm 12.5$ | $32.0 \pm 5.7$ | $62.8 \pm 1.6$ | $53.0 \pm 9.1$ |
|  | P-IQL | $51.2 \pm 5.3$ | $62.5 \pm 4.9$ | $\mathbf{32.0 \pm 3.5}$ | $41.8 \pm 3.8$ | $67.0 \pm 3.0$ | $\mathbf{59.3 \pm 3.7}$ |
|  | CPL | $1.2 \pm 0.8$ | $49.7 \pm 3.0$ | $17.3 \pm 2.5$ | $26.0 \pm 2.2$ | $59.2 \pm 7.7$ | $51.2 \pm 3.0$ |
|  | PPL | $\mathbf{59.7 \pm 18.6}$ | $\mathbf{73.8 \pm 3.3}$ | $25.8 \pm 2.0$ | $\mathbf{58.5 \pm 3.8}$ | $\mathbf{69.8 \pm 2.3}$ | $57.3 \pm 8.6$ |
| Heterogeneous Sparse | SFT | $12.2 \pm 1.0$ | $63.7 \pm 4.7$ | $17.8 \pm 0.8$ | $38.7 \pm 3.0$ | $70.7 \pm 3.8$ | $60.7 \pm 2.5$ |
|  | P-IQL | $48.0 \pm 5.6$ | $71.0 \pm 6.6$ | $\mathbf{44.1 \pm 3.2}$ | $47.5 \pm 3.0$ | $\mathbf{72.0 \pm 4.0}$ | $\mathbf{64.3 \pm 1.0}$ |
|  | CPL | $18.0 \pm 6.1$ | $50.8 \pm 0.8$ | $18.5 \pm 3.0$ | $32.1 \pm 1.6$ | $67.3 \pm 5.5$ | $55.5 \pm 3.3$ |
|  | PPL | $\mathbf{83.8 \pm 3.8}$ | $\mathbf{83.5 \pm 1.8}$ | $34.3 \pm 7.6$ | $\mathbf{60.8 \pm 7.3}$ | $71.2 \pm 1.9$ | $63.3 \pm 4.2$ |

the original DPO formulation, i.e., *forward KL-constrained RLHF implicitly minimizes regret.*

## 4. Experiments

In our experiments, we aim to answer the following questions: (1) Can PPL effectively learn in offline settings composed of heterogeneous data generated by diverse policies? (2) Does incorporating policy labels improve learning performance? (3) Can PPL be effectively applied to online RLHF algorithm? A full report for each question is provided in the Appendix F, G and H.

### 4.1. Experimental Setup

For a fair comparison, we first evaluate the performance of PPL on six robotic manipulation tasks in MetaWorld (Yu et al., 2020), using the same rollout data provided by Hejna et al. (2023). Results from the reproducibility check are included in Appendix E.3. To evaluate performance on offline datasets generated from diverse policies, we aimed to follow CPL's preference dataset generation procedure. However, there are two key differences in our implementation of the critic. First, we utilize raw rollout data without any trajectory truncation. Second, whereas CPL applies a specific technique to reduce TD-error by re-training the critic with all rollout data added to the replay buffer, we generated preference labels without such retraining. As a result, our labels may be noisier than those in CPL. Nevertheless, to ensure a fair comparison, all algorithms were trained using the same set of labels. For further details, please see Appendix E.4.

**Baselines.** We consider CPL as our primary baseline, where the key distinction between PPL and CPL lies in whether the label of the behavior policy is utilized. For ad-

ditional baselines, we include supervised fine-tuning (**SFT**) and Preference-based Implicit $Q$-Learning (**P-IQL**). Specifically, SFT first trains a policy via behavior cloning on all preferred segments in the preference dataset. P-IQL (Hejna & Sadigh, 2024) is a reward-based RLHF algorithm that first learns a reward function from preference data and then derives an optimal policy using the Implicit Q-Learning (IQL) algorithm (Kostrikov et al., 2021). Notably, P-IQL is expected to achieve higher performance, as it not only learns a policy but also simultaneously optimizes a reward function, $Q$-function, and value function.

**Implementation Details.** To generate preference queries without human supervision, we pretrain an SAC model as an oracle that achieves a 100% success rate. Using this pretrained model as a critic, we uniformly sampled segments of length 64 and assigned labels based on estimated regret. To evaluate performance in heterogeneous datasets, we further construct an additional offline dataset by rolling out suboptimal policies with 20% and 50% success rates and combining them. For preference datasets, we conduct experiments under two settings: `Dense`, where comparisons are made between all segment pairs, and `Sparse`, where only one comparison is made per segment.

### 4.2. Can PPL be effectively trained on both homogeneous/heterogeneous offline dataset?

In the previous works, the evaluation of offline datasets has been conducted under homogeneous conditions. However, in practice, offline datasets are more commonly generated by a multiple different policies. Thus, we investigate the following question:

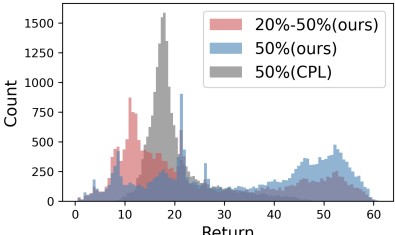

Figure 4: Distribution of returns in homogeneous vs heterogeneous offline dataset in `Button-Press-v2`.

*How would PPL and the baselines perform if the offline dataset were heterogeneous?*

To investigate this, we examine the distribution of segment returns for both types of datasets, as shown in Figure 4. Compared to the homogeneous dataset, the heterogeneous dataset includes rollout data from a policy with a 20% success rate, leading to a higher density of lower-return segments.

In Table 2, we report the impact of diverse behavior policies on performance. PPL consistently outperforms other methods across various dataset conditions in the MetaWorld benchmark, particularly in challenging scenarios with preference sparsity and policy diversity. Interestingly, unlike baseline algorithms, PPL achieves higher performance in `Sparse` settings compared to `Dense` settings. This implies that PPL benefits more from datasets with broader state-action coverage rather than relying on dense pairwise comparisons across all segments. Furthermore, PPL exhibits greater robustness in heterogeneous datasets, outperforming or matching P-IQL despite utilizing only about 6.3% of its parameters. This highlights PPL as an efficient algorithm that maintains strong performance while incurring lower computational costs.

One possible explanation for CPL's lower performance on our dataset is the absence of the *retraining* technique to reduce TD-error—a method uniquely applied within CPL and not commonly adopted in standard practice. However, since all algorithms were trained using the same labels, we attribute this performance gap primarily to CPL's sensitivity to label noise. This sensitivity appears to arise from an implicit assumption within CPL that all training trajectories are generated by an optimal policy.

### 4.3. Does incorporating policy labels improve learning performance?

In this experiment, we examine how the presence and accuracy of policy labels affect performance. Since the offline dataset are fixed and behavior policies are typically unknown, we ablate a pseudo-label setting, assuming each segment was executed deterministically based on the observed actions. Specifically, we introduce **PPL-deterministic**, where the behavior policy for each segment

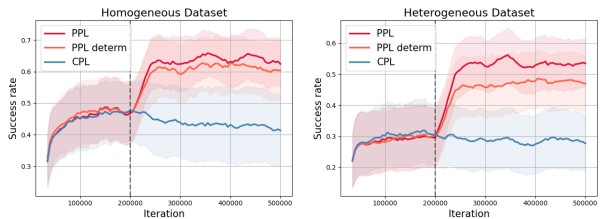

Figure 5: Ablation on deterministic pseudo-labeling. We compare the average performance of PPL and PPL-deterministic across six environments in MetaWorld. The dashed line indicates the point where BC pretraining stops.

is assumed to be fully deterministic (See Lines 4-5 of Algorithm 1). We then compare its performance with PPL.

As shown in Figure 5, comparing PPL with CPL reveals that when behavior policy information is not incorporated into learning, distinguishing environmental stochasticity from behavior policy suboptimality becomes more difficult, resulting in a significant performance gap. As an alternative, using deterministic pseudo-labels for training on offline data without policy labels proves to be a viable approach in homogeneous datasets, causing only a slight performance drop. However, in heterogeneous datasets, their effectiveness decreases, leading to a substantial performance gap. This result suggests that as the dataset becomes more diverse in behavior policies, incorporating policy labels into learning becomes increasingly important.

### 4.4. Can PPL be effectively applied to an online RLHF algorithm?

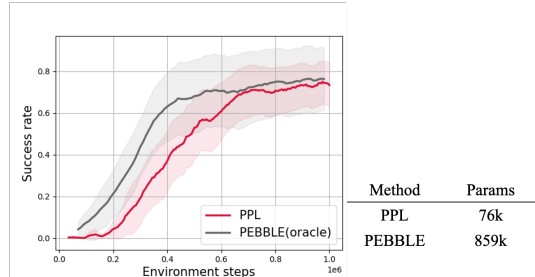

Figure 6: Online learning curves across five MetaWorld tasks, comparing PPL and PEBBLE.

In the online setting, rollouts are directly executed, providing explicit access to policy labels. Leveraging this advantage, we conducted experiments to evaluate whether PPL can effectively serve as an online DPO algorithm. The experiments were conducted *from scratch*, without any pretraining. Unlike the offline setting, we did not apply the asymmetric regularizer in Eq. 1, as out-of-distribution issues were mitigated by the iterative data collection process. We used PEBBLE (Lee et al., 2021) as an oracle because it employs a learned policy trained with unsupervised pre-

training, which accelerates learning. Further implementation details are provided in Appendix E.5.

Figure 6 illustrates the average success rates across five MetaWorld tasks. Notably, despite learning from scratch, the online version of PPL achieves performance comparable to PEBBLE, which leverages unsupervised pretraining. Furthermore, since PPL does not require learning a reward model or a critic, it uses only 8.8% of the parameters compared to PEBBLE, yet still achieves comparable performance. This demonstrates that PPL can serve as a highly efficient online RLHF algorithm.

## 5. Conclusions

In this work, we introduced PPL, a novel DPO framework that incorporates information from the behavior policy through regret-based modeling. We highlighted the issue of likelihood mismatch and addressed it by proposing contrastive KL regularization. Furthermore, we theoretically established that minimizing regret is fundamentally equivalent to optimizing the forward KL-constrained RLHF problem. Empirically, PPL demonstrated strong performance across offline datasets containing rollouts from diverse policies, showcasing its robustness to dataset variations. In online setting, policy labels can be obtained more easily than in the offline case, and PPL effectively learned as an online DPO algorithm. However, we observed that online RLHF method is quite sensitive to the sampling of queries from preference data, suggesting that a more refined analysis is needed for future research.

## Acknowledgements

We would like to thank LG AI Research (Youngsoo Jang, Geonhyeong Kim, Yujin Kim, and Moontae Lee) for their valuable feedback and for providing GPU resources that supported parts of this research. We are grateful to Joey Hejna for sharing implementation details regarding CPL.

This work is in part supported by the National Research Foundation of Korea (NRF, RS-2024-00451435(15%), RS-2024-00413957(15%), RS-2023-00211357(10%)), Institute of Information & communications Technology Planning & Evaluation (IITP, RS-2021-II212068(10%), RS-2025-02305453(15%), RS-2025-02273157(15%), 2021-0-00180(10%), Artificial Intelligence Graduate School Program (Seoul National University, RS-2021-II211343)(10%)) grant funded by the Ministry of Science and ICT (MSIT), Institute of New Media and Communications(INMAC), and the BK21 FOUR program of the Education and Research Program for Future ICT Pioneers, Seoul National University in 2025.

## Impact Statement

This paper presents work whose goal is to advance the field of Machine Learning. There are many potential societal consequences of our work, none which we feel must be specifically highlighted here.

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

## A. Main Proof

**Lemma** (3.2). *(Structural Condition for $\alpha$-optimality) A reward function and a soft optimal $Q$-function where $\pi^*(\cdot|s)$ is $\alpha$-optimal have a one-to-one correspondence with a state-dependent function $\beta : \mathcal{S} \to \mathbb{R}$ as follows,*

$$\mathcal{R}_{\alpha,\pi^*} = \{r_*(s,a) = \alpha \log \pi^*(a|s) + \beta(s) - \gamma \mathbb{E}_{\mathbb{P}}[\beta(s')], \, \forall s \in \mathcal{S}, a \in \mathcal{A}| \, \alpha \geq 0, \beta : \mathcal{S} \to \mathbb{R}\}$$

$$\mathcal{Q}^{\pi^*}_{\alpha,\pi^*} = \{Q^{\pi^*}_*(s,a) = \alpha \log \pi^*(a|s) + \beta(s), \, \forall s \in \mathcal{S}, a \in \mathcal{A}| \, \alpha \geq 0, \beta : \mathcal{S} \to \mathbb{R}\}$$

*Proof.* ($\pi^*$ is $\alpha$-optimal $\Longleftrightarrow Q^{\pi^*}_*(s,a) = \alpha \log \pi^*(a|s) + \beta(s)$ for some $\beta : \mathcal{S} \to \mathbb{R}$.)

Remark that the policy $\pi^*$ is $\alpha$-optimal, if and only if there exists the optimal soft $Q$-function satisfies the following relation:

$$\pi^*(a|s) = \exp\left(\frac{1}{\alpha}(Q^{\pi^*}(s,a) - V^{\pi^*}(s))\right), \quad V^{\pi^*}(s) = \alpha \log \int_{a \in \mathcal{A}} \exp\left(\frac{1}{\alpha}Q^{\pi^*}(s,a)da\right).$$

Since $V^{\pi^*}$ is merely a partition function, letting $X(s,a) = \exp\left(\frac{1}{\alpha}Q^{\pi^*}(s,a)\right)$, we can derive

$$\pi^*(a|s) = \frac{X(s,a)}{\int_{a \in \mathcal{A}} X(s,a)da} \Longleftrightarrow X(s,a) = d(s)\pi^*(a|s) \text{ for some } d : \mathcal{S} \to \mathbb{R}$$

$$\Longleftrightarrow Q^{\pi^*}(s,a) = \alpha \log \pi^*(a|s) + \beta(s) \text{ for some } \beta : \mathcal{S} \to \mathbb{R},$$

where $\beta$ is defined as $\beta(s) = \log d(s)$.

Using the soft Bellman equation, consider a reward function for any state-dependent function $\beta : \mathcal{S} \to \mathbb{R}$ and substitute the expression of $Q^{\pi^*}(s,a)$. Then, we have:

$$r(s,a) := Q^{\pi^*}(s,a) - \gamma \mathbb{E}_{\mathbb{P}}[V^{\pi^*}(s')]$$

$$= \alpha \left( \log \pi^*(a|s) + \beta(s) - \gamma \mathbb{E}_{\mathbb{P}}[\beta(s')] \right)$$

where $\pi^*$ and $\mathbb{P}$ are given. By the definition of optimal soft $Q$-function, we recursively substitute the soft Bellman equation and sum over timesteps:

$$Q^{\pi^*}(s,a) = r(s,a) + \mathbb{E}_{\tau \sim \mathbb{P}^{\pi^*}}\left[\sum_{t>0} \gamma^t(r(s_t,a_t) + \alpha \mathcal{H}^{\pi^*}(\cdot|s_t)) \Big| s_0 = s, a_0 = a\right]$$

$$= \alpha \log \pi^*(a|s) + \beta(s) - \gamma \mathbb{E}_{\mathbb{P}}[\beta(s_1)]$$

$$+ \mathbb{E}_{\tau \sim \mathbb{P}^{\pi^*}}\left[\sum_{t>0} \gamma^t(\beta(s_t) - \gamma \mathbb{E}_{\mathbb{P}}[\beta(s_{t+1})]) \Big| s_0 = s, a_0 = a\right]$$

$$= \alpha \log \pi^*(a|s) + \beta(s) - \gamma^2 \mathbb{E}_{\tau \sim \mathbb{P}^{\pi^*}}[\beta(s_2)]$$

$$+ \mathbb{E}_{\tau \sim \mathbb{P}^{\pi^*}}\left[\sum_{t>1} \gamma^t(\beta(s_t) - \gamma \mathbb{E}_{\mathbb{P}}[\beta(s_{t+1})]) \Big| s_0 = s, a_0 = a\right]$$

$$\vdots$$

$$= \alpha \log \pi^*(a|s) + \beta(s).$$

∎

**Lemma** (3.3). *(Unique Fixed Point of Soft Bellman $\pi$-operator) Let $\pi^*$ is $\alpha$-optimal. For a given policy $\pi$ and $Q$-function $Q^\pi_A \in \mathcal{Q}^\pi$ for any $(s,a) \in \mathcal{S} \times \mathcal{A}$, define the Bellman $\pi$-operator $\mathcal{T}^\pi_* : \mathcal{Q}^\pi \to \mathcal{Q}^\pi$ where*

$$\mathcal{T}^\pi_* Q^\pi_A(s,a) := Q^{\pi^*}_*(s,a) - \gamma \mathbb{E}_{\mathbb{P}}\left[\alpha\left(\mathcal{H}^{\pi^*}(\cdot|s') - \mathcal{H}^\pi(\cdot|s')\right)\right.$$

$$\left. + \mathbb{E}_{\pi^*}[Q^{\pi^*}_*(s',a')] - \mathbb{E}_\pi[Q^\pi_A(s',a')]\right].$$

*Then, $\mathcal{T}^\pi_*$ has a unique fixed point $Q^\pi_*$.*

*Proof.* Consider $Q_A^\pi, Q_B^\pi \in \mathcal{Q}^\pi$. Then

$$\sup_{s,a} \left| \mathcal{T}_*^\pi Q_A^\pi(s,a) - \mathcal{T}_*^\pi Q_B^\pi(s,a) \right| \leq \sup_{s,a} \left| \gamma \mathbb{E}_\mathbb{P} \left[ \mathbb{E}_\pi[Q_A^\pi(s',a')] - \mathbb{E}_\pi[Q_B^\pi(s',a')] \right] \right|$$

$$= \gamma \sup_{s',a'} \left| Q_A^\pi(s',a') - Q_B^\pi(s',a') \right|$$

Hence, $\mathcal{T}_*^\pi$ is a $\gamma$-contraction for any $Q_A^\pi, Q_B^\pi \in \mathcal{Q}^\pi$. Since $\mathcal{Q}^\pi$ is a complete metric space, by using Banach fixed point theorem, $\mathcal{T}_*^\pi$ has a unique fixed point.

Notice that $Q_*^{\pi^*}$ and $Q_*^\pi$ satisfies soft Bellman equation respectively, i.e.,

$$Q_*^{\pi^*}(s,a) = \mathbb{E}_\mathbb{P}\left[ r_*(s,a) + \gamma \mathbb{E}_{\pi^*}[Q_*^{\pi^*}(s',a') + \alpha \mathcal{H}^{\pi^*}(\cdot|s')] \right],$$

$$Q_*^\pi(s,a) = \mathbb{E}_\mathbb{P}\left[ r_*(s,a) + \gamma \mathbb{E}_\pi[Q_*^\pi(s',a') + \alpha \mathcal{H}^\pi(\cdot|s')] \right] \quad \forall (s,a) \in \mathcal{S} \times \mathcal{A}.$$

Then,

$$\mathcal{T}_*^\pi Q_*^\pi(s,a)$$
$$= Q_*^{\pi^*}(s,a) - \gamma \mathbb{E}_\mathbb{P}\left[ \alpha \left( \mathcal{H}^{\pi^*}(\cdot|s') - \mathcal{H}^\pi(\cdot|s') \right) + \mathbb{E}_{\pi^*}[Q_*^{\pi^*}(s',a')] - \mathbb{E}_\pi[Q_*^\pi(s',a')] \right]$$
$$= Q_*^{\pi^*}(s,a) - \gamma \mathbb{E}_\mathbb{P}\left[ \alpha \mathcal{H}^{\pi^*}(\cdot|s') + \mathbb{E}_{\pi^*}[Q_*^{\pi^*}(s',a')] - \left( \alpha \mathcal{H}^\pi(\cdot|s') + \mathbb{E}_\pi[Q_*^\pi(s',a')] \right) \right]$$
$$= \mathbb{E}_\mathbb{P}\left[ r_*(s,a) + \gamma \mathbb{E}_\pi[Q_*^\pi(s',a') + \alpha \mathcal{H}^\pi(\cdot|s')] \right]$$
$$= Q_*^\pi(s,a) \quad \forall (s,a) \in \mathcal{S} \times \mathcal{A}.$$

Hence, $Q_*^\pi$ is a unique fixed point of $\mathcal{T}_*^\pi$. ∎

**Theorem** (3.4). *If a policy $\pi^*$ is $\alpha$-optimal, then for any policy $\pi$,*

$$Q_*^{\pi^*}(s,a) - Q_*^\pi(s,a) = \alpha \bar{D}_{KL}(\pi||\pi^*; s,a)$$

*where the **sequential forward KL divergence** is defined as*

$$\bar{D}_{KL}(\pi||\pi'; s,a) := \mathbb{E}_{\tau \sim \mathbb{P}_{s,a}^\pi}\left[ \sum_{l>0} \gamma^l D_{KL}(\pi(\cdot|s_l)||\pi'(\cdot|s_l)) \right].$$

*Here, $\mathbb{P}_{s,a}^\pi$ is the distribution of trajectories $\tau = (s_0, a_0, \cdots, s_l, a_l, \cdots)$ generated by policy $\pi$ and the transition $\mathbb{P}$, starting at $(s_0, a_0) = (s,a)$.*

*Proof.* Let $\tilde{Q}_*^\pi(s,a) = Q_*^{\pi^*}(s,a) - \alpha \sum_{t>0} \gamma^t \mathbb{E}_{\tau \sim \mathbb{P}^\pi}\left[ D_{KL}(\pi(\cdot|s_t)||\pi^*(\cdot|s_t)) \Big| s_0 = s, a_0 = a \right]$ for all $(s,a) \in \mathcal{S} \times \mathcal{A}$. Then

$$\mathcal{T}_*^\pi \tilde{Q}_*^\pi(s,a)$$
$$= Q_*^{\pi^*}(s,a) - \gamma \mathbb{E}_\mathbb{P}\left[ \alpha \left( \mathcal{H}^{\pi^*}(\cdot|s') - \mathcal{H}^\pi(\cdot|s') \right) + \mathbb{E}_{\pi^*}[Q_*^{\pi^*}(s',a')] - \mathbb{E}_\pi[\tilde{Q}_*^\pi(s',a')] \right]$$
$$= Q_*^{\pi^*}(s,a) - \gamma \mathbb{E}_\mathbb{P}\Bigg[ \alpha \left( \mathcal{H}^{\pi^*}(\cdot|s') - \mathcal{H}^\pi(\cdot|s') \right) + \mathbb{E}_{\pi^*}[\alpha \log \pi^*(s',a') + \beta(s')]$$
$$- \mathbb{E}_\pi\left[ Q_*^{\pi^*}(s',a') - \alpha \sum_{t>0} \gamma^t \mathbb{E}_{\tau \sim \mathbb{P}^\pi}\left[ D_{KL}(\pi(\cdot|s_t)||\pi^*(\cdot|s_t)) \right] \Big| s_1 = s', a_1 = a' \right] \Bigg]$$
$$= Q_*^{\pi^*}(s,a) - \gamma \mathbb{E}_\mathbb{P}\Bigg[ \beta(s') - \alpha \mathcal{H}^\pi(\cdot|s') - \mathbb{E}_\pi[Q_*^{\pi^*}(s',a')]$$
$$+ \alpha \mathbb{E}_\pi\left[ \sum_{t>0} \gamma^t \mathbb{E}_{\tau \sim \mathbb{P}^\pi}\left[ D_{KL}(\pi(\cdot|s_t)||\pi^*(\cdot|s_t)) \right] \Big| s_1 = s', a_1 = a' \right] \Bigg]$$

$$= Q_*^{\pi^*}(s,a) - \alpha\gamma\mathbb{E}_{\mathbb{P}}\Big[D_{KL}(\pi(\cdot|s')||\pi^*(\cdot|s'))\Big] - \alpha\sum_{t>1}\gamma^t\mathbb{E}_{\tau\sim\mathbb{P}^\pi}\Big[D_{KL}(\pi(\cdot|s_t)||\pi^*(\cdot|s_t))\Big|s_0=s,a_0=a\Big]$$

$$= Q_*^{\pi^*}(s,a) - \alpha\sum_{t>0}\gamma^t\mathbb{E}_{\tau\sim\mathbb{P}^\pi}\Big[D_{KL}(\pi(\cdot|s_t)||\pi^*(\cdot|s_t))\Big|s_0=s,a_0=a\Big]$$

$$= \tilde{Q}_*^\pi(s,a)$$

which implies that $\tilde{Q}_*^\pi$ is a unique fixed point of $\mathcal{T}_*^\pi$. In Lemma 3.3, we observe that $\mathcal{T}_*^\pi$ has a unique fixed point $Q_*^\pi$. Hence,

$$Q_*^\pi(s,a) = Q_*^{\pi^*}(s,a) - \alpha\sum_{t>0}\gamma^t\mathbb{E}_{\tau\sim\mathbb{P}^\pi}\Big[D_{KL}(\pi(\cdot|s_t)||\pi^*(\cdot|s_t))\Big|s_0=s,a_0=a\Big]$$

∎

## B. Further Theoretical Analysis & Discussion

### B.1. Mathematical derivation of PPL framework

We recall the PPL model and objective:

$$P_{\pi_\psi}^{(\pi^+,\pi^-)}[\zeta^+\succ\zeta^-] = \sigma\Bigg(-\sum_{t\geq0}\text{Reg}_{\pi_\psi}^{\pi^+}(s_t^+,a_t^+) - \text{Reg}_{\pi_\psi}^{\pi^-}(s_t^-,a_t^-)\Bigg),$$

$$\mathcal{L}_{\text{PPL}}(\pi_\psi;\mathcal{D}) = -\mathbb{E}_{(\zeta^+,\zeta^-,y,p)\sim\mathcal{D}}\Bigg[\log\sigma\Bigg(-\sum_{t\geq0}\text{Reg}_{\pi_\psi}^{\pi^+}(s_t^+,a_t^+) - \text{Reg}_{\pi_\psi}^{\pi^-}(s_t^-,a_t^-)\Bigg)\Bigg]$$

where

$$-\text{Reg}_{\pi^*}^\pi(s_t,a_t) := -(V_*^{\pi^*}(s_t) - Q_*^\pi(s_t,a_t)).$$

Here, a negative regret at $(s_t,a_t)$ can be decomposed into two components:

$$-\text{Reg}_{\pi^*}^\pi(s_t,a_t) = \alpha\Bigg(\underbrace{\log\pi^*(a_t|s_t)}_{\text{increase likelihood}} - \underbrace{\mathbb{E}_{\tau\sim\mathbb{P}_{s_t,a_t}^\pi}\Big[\sum_{l>0}\gamma^l D_{\text{KL}}(\pi(\cdot|s_l)||\pi^*(\cdot|s_l))\Big]}_{\text{decrease sequential forward KL divergence}}\Bigg)$$

*Proof.* By the definition of regret,

$$-\text{Reg}_{\pi^*}^\pi(s_t,a_t) := -(V_*^{\pi^*}(s_t) - Q_*^\pi(s_t,a_t)).$$

$$= -\Big(\mathbb{E}_{\pi^*}[Q_*^{\pi^*}(s_t,a) - \alpha\log\pi^*(\cdot|s_t)]\Big) + Q_*^{\pi^*}(s_t,a_t) - \alpha\bar{D}_{KL}(\pi||\pi^*;s_t,a_t)$$

$$= \cancel{-\beta(s_t)} + \alpha\log\pi^*(a_t|s_t) + \cancel{\beta(s_t)} - \alpha\bar{D}_{KL}(\pi||\pi^*;s_t,a_t)$$

$$= \alpha\Big(\log\pi^*(a_t|s_t) - \bar{D}_{KL}(\pi||\pi^*;s_t,a_t)\Big) \tag{5}$$

∎

### B.2. Regret is invariant under policy-invariant transformations (Corollary 3.5)

As noted in Lemma 3.2, any policy-invariant transformation can be expressed as a combination of a state-dependent function $\beta(s)$ and a scaled log-likelihood term $\alpha\log\pi(a|s)$, where $\alpha$ represents the temperature parameter in the MaxEnt framework. Specifically, for any transformation of the reward function that preserves the optimal policy, we can rewrite the modified reward as:

$$r(s,a) = \alpha\log\pi(a|s) + \beta(s).$$

This formulation extends the classical reward shaping result of Ng et al. (1999) by explicitly incorporating the policy-dependent term $\alpha \log \pi(a|s)$, which accounts for transformations in the likelihood space. This insight allows us to generalize policy-invariant transformations and directly integrate them into preference-based learning objectives.

Using this representation, we can reformulate the sequential DPO objective with a policy-invariant transformation as follows:

$$\mathcal{L}_{\text{DPO}(\beta)}(\pi_\psi; \mathcal{D}) = -\mathbb{E}_{\mathcal{D}} \Big[ \log \sigma \Big( \sum_{t \geq 0} \Big\{ \log \frac{\pi_\psi(a_t^+|s_t^+)}{\pi_{\text{ref}}(a_t^+|s_t^+)} + \beta(s_t^+) - \gamma \mathbb{E}_{s_t' \sim \mathbb{P}(\cdot|s_t^+, a_t^+)}[\beta(s_t')] \Big\}$$
$$- \Big\{ \log \frac{\pi_\psi(a_t^-|s_t^-)}{\pi_{\text{ref}}(a_t^-|s_t^-)} + \beta(s_t^+) - \gamma \mathbb{E}_{s_t' \sim \mathbb{P}(\cdot|s_t^-, a_t^-)}[\beta(s_t')] \Big\} \Big) \Big]. \tag{6}$$

The existence of multiple objectives that preserve the optimal policy through reward shaping has been explored in previous work, particularly in the *variance reduction* schemes of policy gradient methods. Schulman et al. (2015) introduced the *generalized advantage estimate* (GAE) as a method to reduce the variance of policy gradient estimates, effectively selecting an appropriate $\beta(s)$ to improve stability and efficiency in learning. Similarly, in Equation 6, the standard DPO framework assumes $\beta(s) = 0$, but optimizing $\beta(s)$ to minimize the variance of gradient estimates could lead to more stable training.

In contrast, as shown in Equation 5, regret-based formulations naturally eliminate $\beta(s)$ by definition, avoiding the challenges associated with policy-invariant transformations. This property ensures that regret serves as a unique and well-defined objective function, making it inherently robust without requiring explicit variance reduction techniques.

### B.3. Reformulating the MaxEnt objective with negative regret as the reward (Theorem 3.6)

**Corollary** (3.6). *Maximizing the MaxEnt objective with negative regret as the reward is equivalent to minimizing the sequential forward KL divergence between the learned policy and the behavior policy for each preferred state-action pair in the dataset, i.e.,*

$$\arg\max_{\pi_\psi} \Big( \mathbb{E}_{\zeta^+ \sim \mathcal{D}}[-Reg_{\pi_\psi}^{\pi^+}(s^+, a^+) - \alpha \log \pi_\psi(a^+|s^+)] \Big)$$
$$\equiv \arg\min_{\pi_\psi} \Big( \mathbb{E}_{\zeta^+ \sim \mathcal{D}}[\bar{D}_{KL}(\pi^+||\pi_\psi; s^+, a^+)] \Big). \tag{7}$$

*Proof.* Consider a dataset $\mathcal{D}$ and a set of sampled preferred segments $\{\zeta_i^+\}_{i=1}^N$ which are generated by behavior policy $\pi_i^+$ respectively. To avoid notation ambiguity, we emphasize that the subscript $i$ in this proof denotes the index of each individual samples. When defining the reward function as the negative regret, the optimal policy of Maxent objective $\pi_{\text{Reg}}^*$ can be reformulated as:

$$\pi_{reg}^* := \arg\max_{\pi_\psi} \Big( \frac{1}{N} \sum_{i=1}^N \Big[ - \text{Reg}_{\pi_\psi}^{\pi_i^+}(s_i^+, a_i^+) - \alpha \log \pi_\psi(a_i^+|s_i^+) \Big] \Big)$$
$$= \arg\max_{\pi_\psi} \Big( \frac{1}{N} \sum_{i=1}^N \Big[ Q_{\pi_\psi}^{\pi_i^+}(s_i^+, a_i^+) - V_{\pi_\psi}^{\pi_\psi}(s_i^+) - \alpha \log \pi_\psi(a_i^+|s_i^+) \Big] \Big)$$
$$= \arg\max_{\pi_\psi} \Big( \frac{1}{N} \sum_{i=1}^N \Big[ \alpha \log \pi_\psi(a_i^+|s_i^+) - \alpha \bar{D}_{KL}(\pi||\pi_\psi; s_i^+, a_i^+) - \alpha \log \pi_\psi(a_i^+|s_i^+) \Big] \Big)$$
$$= \arg\min_{\pi_\psi} \Big( \frac{1}{N} \sum_{i=1}^N \bar{D}_{KL}(\pi_i^+||\pi_\psi; s_i^+, a_i^+) \Big)$$

$\blacksquare$

Notably, the minimum is achieved if and only if $\pi_\psi(a_i^+|s_i^+) = \pi(a_i^+|s_i^+)$ for each $i \in [N]$. This formulation demonstrates that maximizing the MaxEnt objective with a regret-based reward is fundamentally equivalent to minimizing the sequential forward KL divergence for each segment.

**Discussion.** The regret-based DPO framework can be reinterpreted as a process that aggregates the behavior policies underlying the given dataset, aligning the learned policy to preferred actions by reducing the sequential forward KL divergence. If, as assumed in CPL, the behavior policies of all preferred segments in dataset $\mathcal{D}$ correspond to the optimal policy $\pi^*$ (or can be constructed as such), then PPL is guaranteed to converge to the optimal policy.

However, in practical RLHF settings, such an assumption rarely holds. Unlike standard reinforcement learning, where an agent maximizes a predefined reward function, RLHF optimizes for policy alignment rather than absolute optimality. In the DPO framework, the reward function is implicitly constructed to make the aligned policy the optimal one within the given preference dataset. As a result, the optimal policy under the learned reward function is already the policy obtained through alignment, making it unnecessary to perform an additional RL algorithm to reach the optimal policy.

To achieve further improvements, it is crucial to expand the dataset by rolling out new policies and incorporating additional preference data. This process enhances dataset coverage while enabling the learned reward function to extrapolate more effectively. Without such iterative expansion, RLHF remains constrained by the limitations of the static dataset, preventing meaningful policy improvements beyond the scope of the initially collected preferences.

## C. Pseudocode

---
**Algorithm 1** Policy-labeled Preference Learning (PPL)
---
**Input:** number of queries $N$, trajectory dataset $\mathcal{E}$, minibatch size $D$

1: Initialize policy parameters $\psi$
2: **for** $n = 1, \cdots, N$ **do**
3:      Sample $\zeta, \zeta' \sim \mathcal{E}$
4:      **if** policy label $\pi(a_t|s_t), \pi(a_t'|s_t')$ unknown **then**
5:          $\pi(\cdot|s_t) \leftarrow \delta_{a_t}\ \pi(\cdot|s_t') \leftarrow \delta_{a_t'}$
6:      **end if**
7:      Label the behavior policy $p = (\pi, \pi')$
8:      Instruct the preference label $y = (y(0), y(1))$
9:      Store preference $\mathcal{D} \leftarrow \mathcal{D} \cup \{(\zeta, \zeta', y, p)\}$                // Create Policy-labeled Preference Queries
10: **end for**
11: **for** $t = 1$ **to** $T$ **do**
12:      Sample minibatch $\{(\zeta, \zeta', y, p)_d\}_{d=1}^{D} \sim \mathcal{D}$
13:      $\psi \leftarrow \arg\min_\psi \mathcal{L}_{\text{PPL}}(\pi_\psi; \mathcal{D})$                // Policy Learning
14: **end for**
---

# D. Variants of PPL and Baselines

**BC:** BC (Behavior Cloning) is the initial stage in RLHF, where the policy is trained to maximize the likelihood of the demonstrated actions given the corresponding states:

$$\mathcal{L}_{\text{BC}}(\pi_\psi; \mathcal{D}) = -\mathbb{E}_{\zeta \sim \mathcal{D}} \Big[ \sum_{t \geq 0} \log \pi_\psi(a_t | s_t) \Big]$$

**SFT:** SFT (Supervised Fine Tuning) is trained to maximize the likelihood of the demonstrated actions given the corresponding states in preferred segments:

$$\mathcal{L}_{\text{SFT}}(\pi_\psi; \mathcal{D}) = -\mathbb{E}_{\zeta^+ \sim \mathcal{D}} \Big[ \sum_{t \geq 0} \log \pi_\psi(a_t^+ | s_t^+) \Big]$$

**CPL:** CPL (Hejna et al., 2023) is our primary baseline, where the optimal advantage is defined as the score function:

$$S_{\text{CPL}}(\pi_\psi; \zeta^+) - S_{\text{CPL}}(\pi_\psi; \zeta^-) = \sum_{t \geq 0} \log \frac{\pi_\psi(a_t^+ | s_t^+)}{\pi_\psi(a_t^- | s_t^-)}.$$

The objective is to minimize the following loss function:

$$\mathcal{L}_{\text{CPL}}(\pi_\psi; \mathcal{D}) = -\mathbb{E}_{(\zeta^+, \zeta^-) \sim \mathcal{D}} \Big[ \log \sigma \big( S_{\text{CPL}}(\pi_\psi; \zeta^+) - S_{\text{CPL}}(\pi_\psi; \zeta^-) \big) \Big]$$

A key issue raised in CPL is assigning high weights to OOD actions while still maintaining the same optimal policy. This leads to extrapolation too much into unseen states, ultimately degrading performance. To mitigate this, an asymmetric regularizer is introduced:

$$S_{\text{CPL}(\lambda)}(\pi_\psi; \zeta^+) - S_{\text{CPL}(\lambda)}(\pi_\psi; \zeta^-) = S_{\text{CPL}}(\pi_\psi; \zeta^+) - \lambda S_{\text{CPL}}(\pi_\psi; \zeta^-) = \sum_{t \geq 0} \log \frac{\pi_\psi(a_t^+ | s_t^+)}{\pi_\psi(a_t^- | s_t^-)^\lambda}$$

**PPL:** Based on policy deviation lemma in Theorem 3.4, PPL extends CPL by incorporating entropy regularization and KL divergence-based constraints, making preference learning more structured. The score function includes multiple terms:

$$S_{\text{PPL}}(\pi_\psi; \zeta^+, \pi^+) - S_{\text{PPL}}(\pi_\psi; \zeta^-, \pi^-)$$
$$= \sum_{t \geq 0} \left[ \log \frac{\pi_\psi(a_t^+ | s_t^+)}{\pi_\psi(a_t^- | s_t^-)} + \frac{1}{L} \sum_{l=1}^{L} \Big( -D_{KL}(\pi^+(\cdot | s_{t+l}^+) || \pi_\psi(\cdot | s_{t+l}^+)) + D_{KL}(\pi^-(\cdot | s_{t+l}^-) || \pi_\psi(\cdot | s_{t+l}^-)) \Big) \right],$$

and the objective function is:

$$\mathcal{L}_{\text{PPL}}(\pi_\psi; \mathcal{D}) = -\mathbb{E}_{(\zeta^+, \zeta^-) \sim \mathcal{D}} \Big[ \log \sigma \big( S_{\text{PPL}}(\pi_\psi; \zeta^+) - S_{\text{PPL}}(\pi_\psi; \zeta^-) \big) \Big]$$

The score function of PPL with the same asymmetric regularizer as CPL is given by:

$$S_{\text{PPL}(\lambda)}(\pi_\psi; \zeta^+, \pi^+) - S_{\text{PPL}(\lambda)}(\pi_\psi; \zeta^-, \pi^-) = S_{\text{PPL}}(\pi_\psi; \zeta^+, \pi^+) - \lambda S_{\text{PPL}}(\pi_\psi; \zeta^-, \pi^-)$$
$$= \sum_{t \geq 0} \left[ \log \frac{\pi_\psi(a_t^+ | s_t^+)}{\pi_\psi(a_t^- | s_t^-)^\lambda} + \frac{1}{L} \sum_{l=1}^{L} \Big( -D_{KL}(\pi^+(\cdot | s_{t+l}^+) || \pi_\psi(\cdot | s_{t+l}^+)) + \lambda D_{KL}(\pi^-(\cdot | s_{t+l}^-) || \pi_\psi(\cdot | s_{t+l}^-)) \Big) \right],$$

**PPL-deterministic:** If policy-label is unknown, we apply deterministic pseudo-labels by assuming that each segment was generated by a deterministic policy that executed the observed action.

$$S_{\text{PPL-d}}(\pi_\psi; \zeta^+) - S_{\text{PPL-d}}(\pi_\psi; \zeta^-) = \sum_{t \geq 0} \left[ \log \frac{\pi_\psi(a_t^+|s_t^+)}{\pi_\psi(a_t^-|s_t^-)} + \frac{1}{L} \sum_{l=1}^{L} \log \frac{\pi_\psi(a_{t+l}^+|s_{t+l}^+)}{\pi_\psi(a_{t+l}^-|s_{t+l}^-)} \right]$$

$$S_{\text{PPL-ref}}(\pi_\psi; \zeta^+) - S_{\text{PPL-ref}}(\pi_\psi; \zeta^-) = \sum_{t \geq 0} \Big( \log \frac{\pi_\psi(a_t^+|s_t^+)}{\pi_{\text{ref}}(a_t^+|s_t^+)} - \log \frac{\pi_\psi(a_t^-|s_t^-)}{\pi_{\text{ref}}(a_t^-|s_t^-)}$$

$$+ \mathcal{H}^{\pi_\psi}(\cdot|s_t^+) - \mathcal{H}^{\pi_\psi}(\cdot|s_t^-) - \sum_{l=1}^{L} D_{KL}(\pi^+(\cdot|s_{t+l}^+)||\pi_\psi(\cdot|s_{t+l}^+)) + \sum_{l=1}^{L} D_{KL}(\pi^-(\cdot|s_{t+l}^-)||\pi_\psi(\cdot|s_{t+l}^-))$$

$$- \mathcal{H}^{\pi_{\text{ref}}}(\cdot|s_t^+) + \mathcal{H}^{\pi_{\text{ref}}}(\cdot|s_t^-) + \sum_{l=1}^{L} D_{KL}(\pi^+(\cdot|s_{t+l}^+)||\pi_{\text{ref}}(\cdot|s_{t+l}^+)) - \sum_{l=1}^{L} D_{KL}(\pi^-(\cdot|s_{t+l}^-)||\pi_{\text{ref}}(\cdot|s_{t+l}^-)) \Big)$$

# E. Implementation Details

## E.1. Hyperparameter Setting

Table 3: Hyperparameter settings for offline implementation.

| Hyperparameter | State |
|---|---|
| Total Training Steps | 500k |
| Pre-training Steps (except P-IQL) | 200k |
| Batch Size | 96 |
| Segment Size | 64 |
| Fixed log std | -1.5 |
| Actor Dropout | 0.0 (0.25 for CPL reproduce) |
| Architecture | [256, 256] MLP Gaussian |

Table 4: Hyperparameters for online implementation

| Hyperparameter | State |
|---|---|
| Total Environment Steps | 1m |
| Segment Size | 32 |
| Fixed log std | -1.0 |
| Query Frequency(steps) | 1000 |
| Policy update Frequency(steps) | 1000 |
| Episode Length | 250 |
| Learning rates | 3e-4 |
| Temperature $\alpha$ | 0.1 |
| Asymmetric regularizer $\lambda$ | 1.0 |
| BC weights | 0 |
| $\gamma$ | 1 |
| Actor Dropout | 0.0 |
| Architecture | [256, 256] MLP Gaussian |

Table 5: Hyperparameters for PPL, CPL, SFT, and P-IQL

| Hyperparameter | PPL | CPL | SFT | P-IQL |
|---|---|---|---|---|
| Learning rates | 1e-4 | 1e-4 | 1e-4 | 1e-4 |
| Temperature $\alpha$ | 0.1 | 0.1 | 0.1 | 0.1 |
| Asymmetric regularizer $\lambda$ | 0.5 | 0.5 | - | - |
| BC weights | 0 | 0 | 0 | 0 |
| $\gamma$ | 1 | 1 | 1 | 1 |
| Number of Parameters | 76k | 76k | 76k | 859k |

### E.2. MetaWorld Benchmark

Our experiments were conducted on six MetaWorld environments: `Bin-Picking, Button-Press, Door-Open, Drawer-Open, Plate-Slide,` and `Sweep-Into`. Each task requires precise control of a robotic arm to interact with objects in a structured environment. The diverse task set includes object relocation, pushing, pulling, and fine-grained manipulation, making it a suitable testbed for reinforcement learning from preference-based feedback.

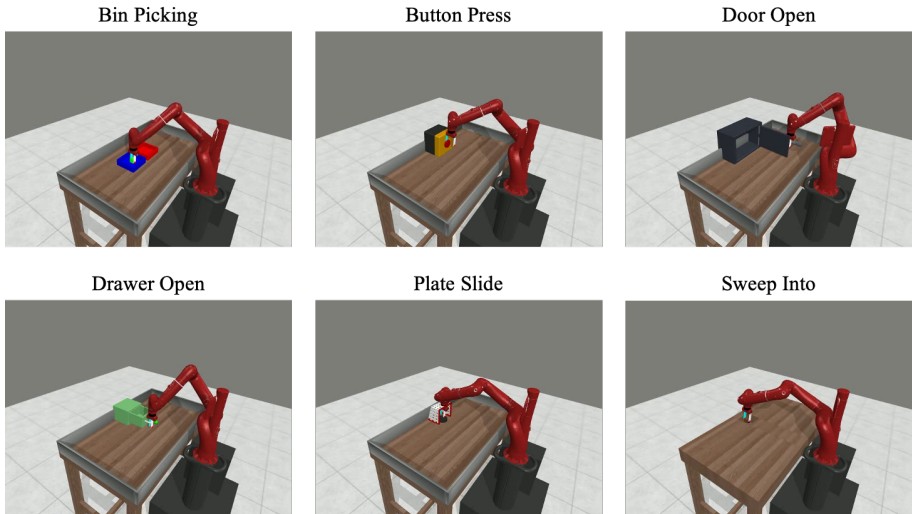

Figure 7: Visualization of the MetaWorld Benchmark Tasks.

Each environment is designed with a handcrafted reward function tailored to its objective. Instead of human annotations, we trained a critic using SAC to assign labels. During our experiments, we observed that return did not always align well with success rates. In case of `Door-Open`, despite achieving the highest return, PPL exhibited a relatively low success rate. This implies that the environment allows reward exploitation due to the imprecise design of the reward function.

### E.3. Reproducibility Check

For a fair comparison, we first verified the reproducibility of CPL using the Metaworld `State Dense` and `State Sparse` datasets provided by Hejna et al. (2023) and evaluated the performance of PPL on these datasets. We used the official CPL implementation (`https://github.com/jhejna/cpl`) without modifications and ensured reproducibility by fixing the random seed ([123, 231, 312, 321]). The figure below presents the PPL performance alongside the reproduced CPL results. The horizontal dashed line represents the scores reported in CPL, confirming the reproducibility of the algorithm. The vertical dashed line indicates the point where behavior cloning (BC) training stops.

In all environments except `Plate-Slide-v2`, the reproduced CPL performance closely matches the reported values, with deviations attributed to seed variability. Across the provided datasets, PPL exhibits comparable overall performance to CPL.

Table 6: Success rates of all methods on six tasks from the MetaWorld across different datasets from Hejna et al. (2023). Each score is reported as the highest average performance across four seeds over a 200-episode evaluation window.

|  |  | Bin Picking | Button Press | Door Open | Drawer Open | Plate Slide | Sweep Into |
|---|---|---|---|---|---|---|---|
| State 2.5k Dense | CPL(Reported) | $80.0 \pm 2.5$ | $24.5 \pm 2.1$ | $80.0 \pm 6.8$ | $83.6 \pm 1.6$ | $61.1 \pm 3.0$ | $70.4 \pm 3.0$ |
|  | CPL(Reproduced) | $76.0 \pm 4.1$ | $24.9 \pm 4.7$ | $75.5 \pm 6.0$ | $87.6 \pm 2.8$ | $45.3 \pm 10.4$ | $74.5 \pm 3.4$ |
|  | PPL | $77.7 \pm 2.6$ | $30.2 \pm 7.8$ | $76.7 \pm 7.1$ | $84.2 \pm 2.4$ | $41.7 \pm 3.2$ | $79.2 \pm 5.5$ |
| State 20k Sparse | CPL(Reported) | $83.2 \pm 3.5$ | $29.8 \pm 1.8$ | $77.9 \pm 9.3$ | $79.1 \pm 5.0$ | $56.4 \pm 3.9$ | $81.2 \pm 1.6$ |
|  | CPL(Reproduced) | $69.1 \pm 21.4$ | $25.5 \pm 5.3$ | $74.4 \pm 3.5$ | $80.9 \pm 4.5$ | $41.1 \pm 4.9$ | $80.5 \pm 2.8$ |
|  | PPL | $83.0 \pm 3.7$ | $25.4 \pm 2.8$ | $72.2 \pm 1.7$ | $79.0 \pm 4.0$ | $42.9 \pm 1.6$ | $76.0 \pm 2.0$ |

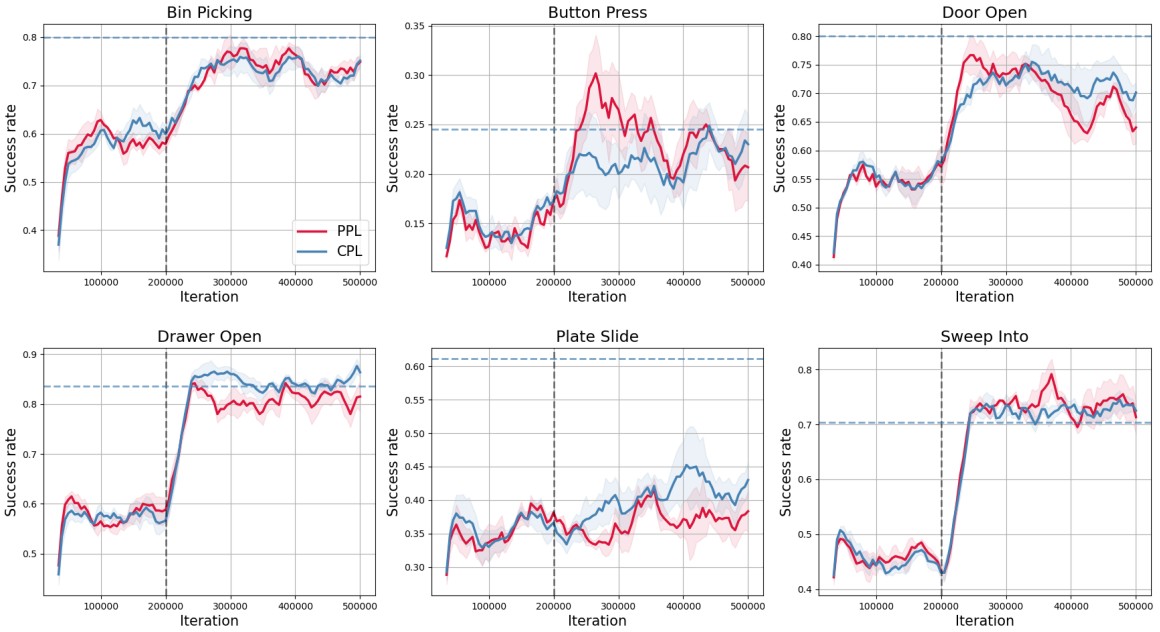

Figure 8: Reproducibility check on `State Dense` dataset

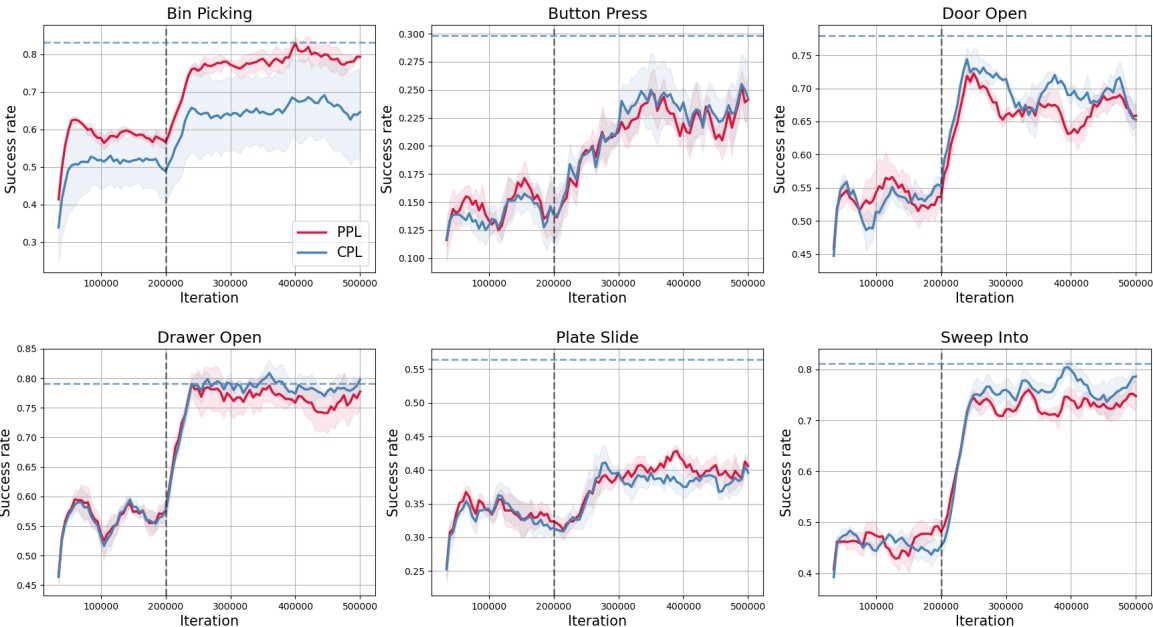

Figure 9: Reproducibility check on `State Sparse` dataset

### E.4. Offline dataset generation and its distribution

We construct a heterogeneous dataset by incorporating various policies, following the dataset generation method used in Hejna et al. (2023). Specifically, we load suboptimal SAC checkpoints with success rates of 20% and 50% using the same approach. During rollouts, we introduce Gaussian noise with a standard deviation of 0.3 and rolling out 20,000 episodes, each lasting 250 steps, using their suboptimal soft actor-critic (SAC) (Haarnoja et al., 2018) checkpoints, which achieved an approximate 50% success rate.

While following this data generation procedure, we found a step in the reference code where transitions following a success signal were explicitly truncated. This truncation was intended to prevent segments from being overly dominated by successful transitions. However, we opted to retain the raw data without truncation. As a result, the distribution of our 50% success rate dataset differs from that of Hejna et al. (2023). To highlight this difference, we provide a visualization of the data distribution across environments.

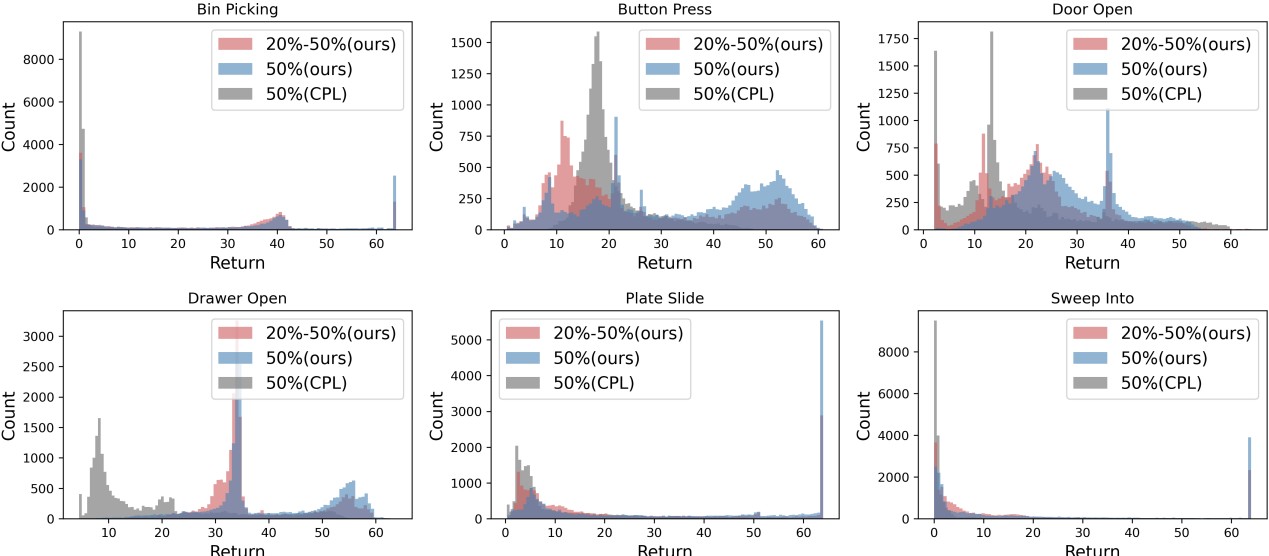

Figure 10: Comparison of return distributions across environments for different dataset configurations. The histograms illustrate the distribution of the partial returns for segments with 20% and 50% success rates generated using our method **(red and blue)** and the 50% success rate dataset from Hejna et al. (2023) **(gray)**.

For our experiments, we generated the following four datasets: `Homogeneous Dense`, `Homogeneous Sparse`, `Hetereogeneous Dense`, and `Heterogeneous Sparse`. In the additional experimental setting, we kept all aspects—such as the hyperparameters of all algorithms, the SAC critic, and the label generation method—identical to the original setup, modifying only the dataset. Interestingly, CPL exhibited significant performance variations depending on the dataset, whereas PPL demonstrated robust performance across diverse datasets. The robustness of PPL's performance can be attributed to its ability to adjust the magnitude of feedback for diverse policies and accurately reflect the likelihood of each segment.

## E.5. Online Implementation

In the online setting, we use a Gaussian actor with a fixed standard deviation to maintain consistency with the offline setting. The model is trained from scratch without any pretraining. The online learning process consists of three phases. First, rollouts are conducted in the environment for a fixed number of steps to generate trajectory data. Next, preference queries and labels are constructed from segments of these trajectories. Finally, the policy is updated using the generated preference query data.

During the rollout phase, actions are sampled from a stochastic policy without additional exploration strategies. In the query generation phase, two policies are selected for comparison, with one always being the most recent and the other randomly chosen from the last 25 policies. Segments from the most recent policy are first over-sampled at three times the required number, then ranked based on their regret scores relative to the current policy. The top-ranked segments are retained, while segments from the other policy are sampled uniformly at random. Preference labels are assigned according to the method described in Appendix D.2 of Hejna et al. (2023).

In the policy update phase, stochastic gradient updates are applied over a fixed number of epochs using all preference query data collected up to that point. Unlike reward-based preference learning methods, which predominantly generate preference queries early in training and subsequently optimize policies using a learned reward function and an RL algorithm, the online PPL algorithm continuously collects preference queries throughout the entire training process. This ensures sustained policy improvement over time.

To reproduce the online baseline PEBBLE algorithm, we utilized the official B-Pref implementation (https://github.com/rll-research/BPref) and adhered to the hyperparameter settings and random seeds reported in the original paper. Our online experiments were performed on five tasks from the MetaWorld benchmark: `Button Press`, `Door Open`, `Drawer Open`, `Plate Slide`, and `Sweep Into`. All hyperparameters were kept consistent across tasks, except for the total number of preference queries, which was set to match the values specified for each environment in the PEBBLE paper.

# F. Experimental Results on Homogeneous/ Heterogeneous Datasets (Section 4.2)

## F.1. `Homogeneous Dense` Offline Dataset

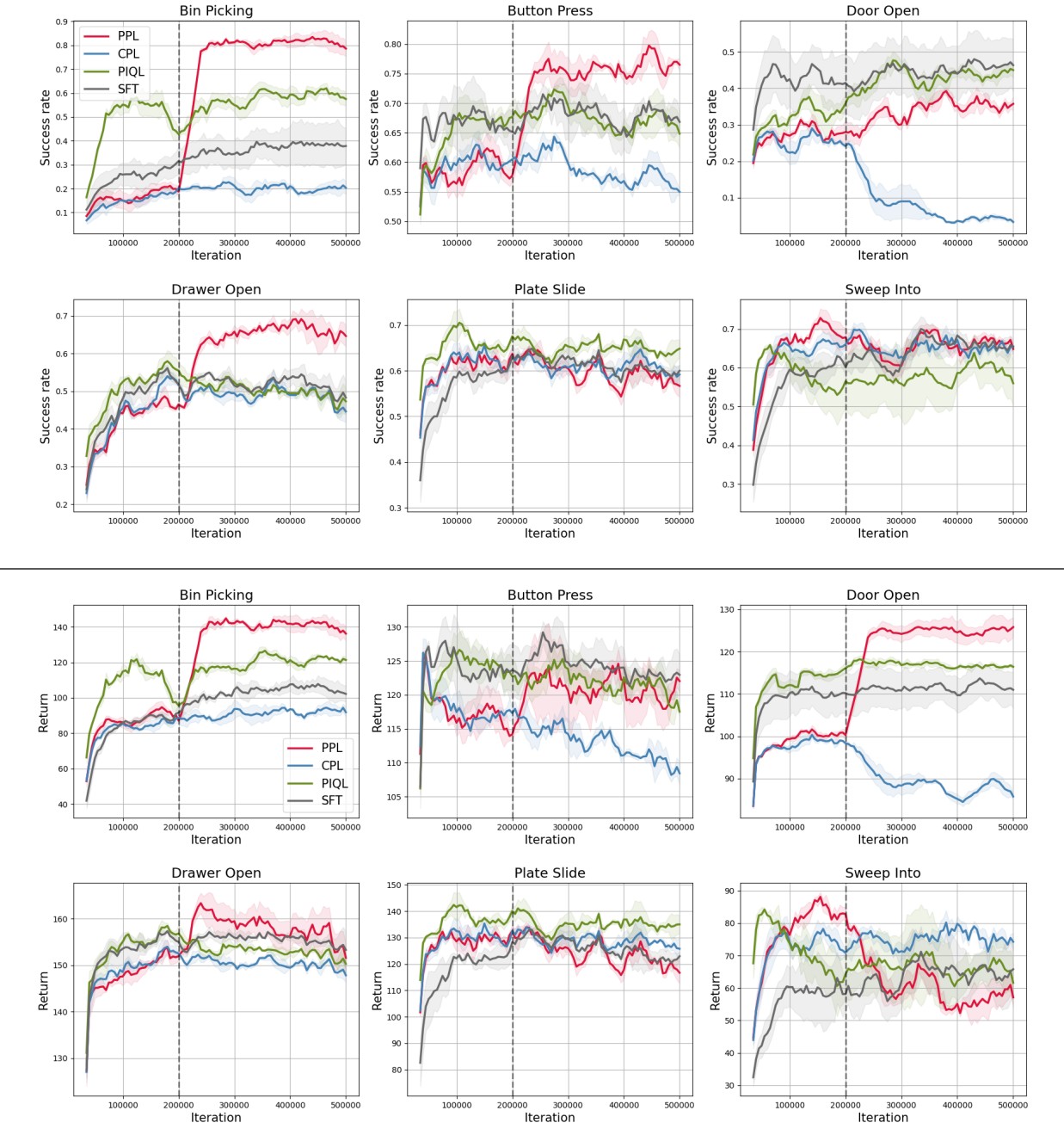

Figure 11: Performance comparison of different methods on the `Homogeneous Dense` dataset across six MetaWorld tasks. The top row shows the success rate over training iterations, while the bottom row presents the corresponding return values.

## F.2. `Homogeneous Sparse` Offline Dataset

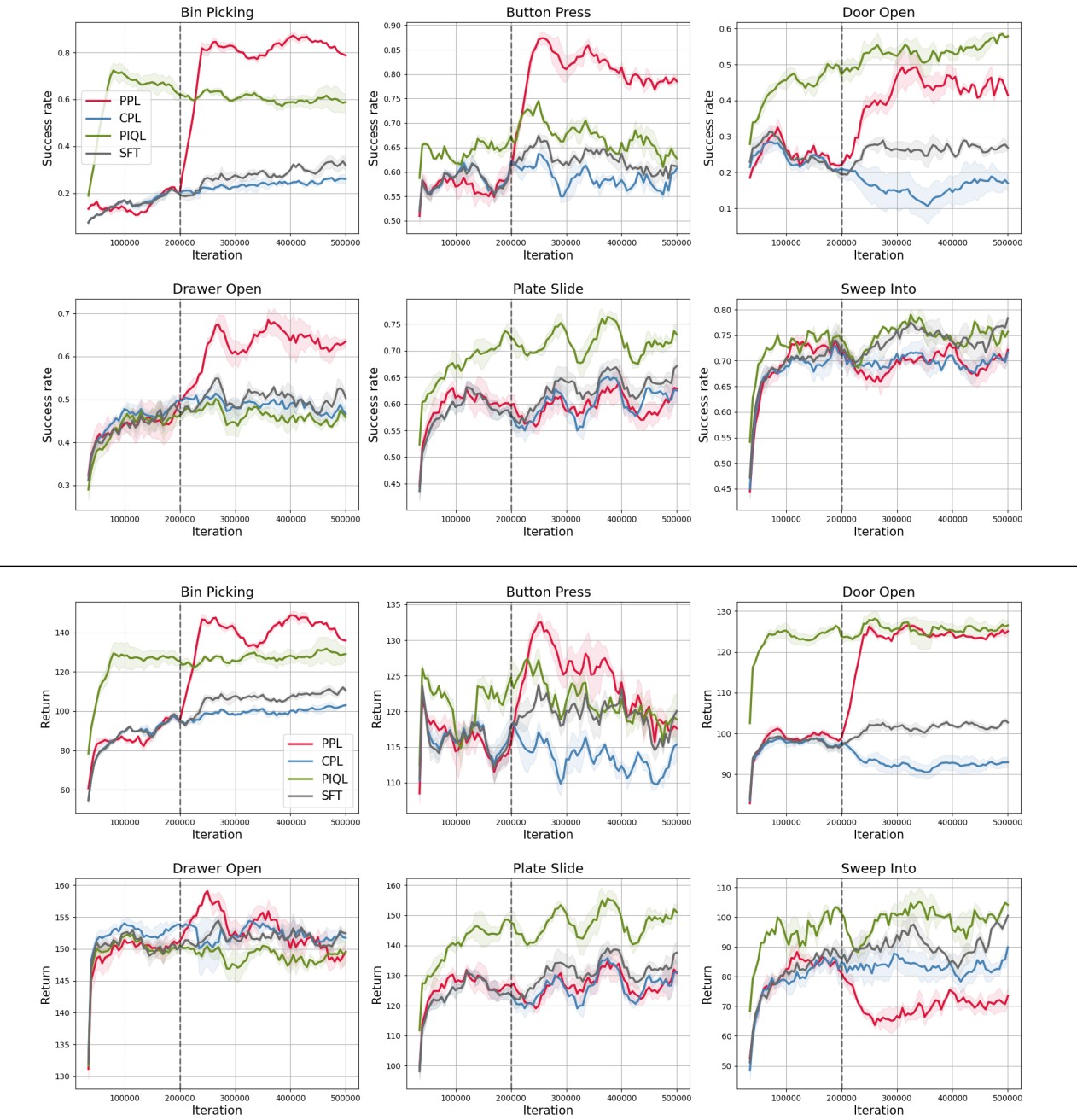

Figure 12: Performance comparison of different methods on the `Homogeneous Sparse` dataset across six MetaWorld tasks.

## F.3. `Heterogeneous Dense` Offline Dataset

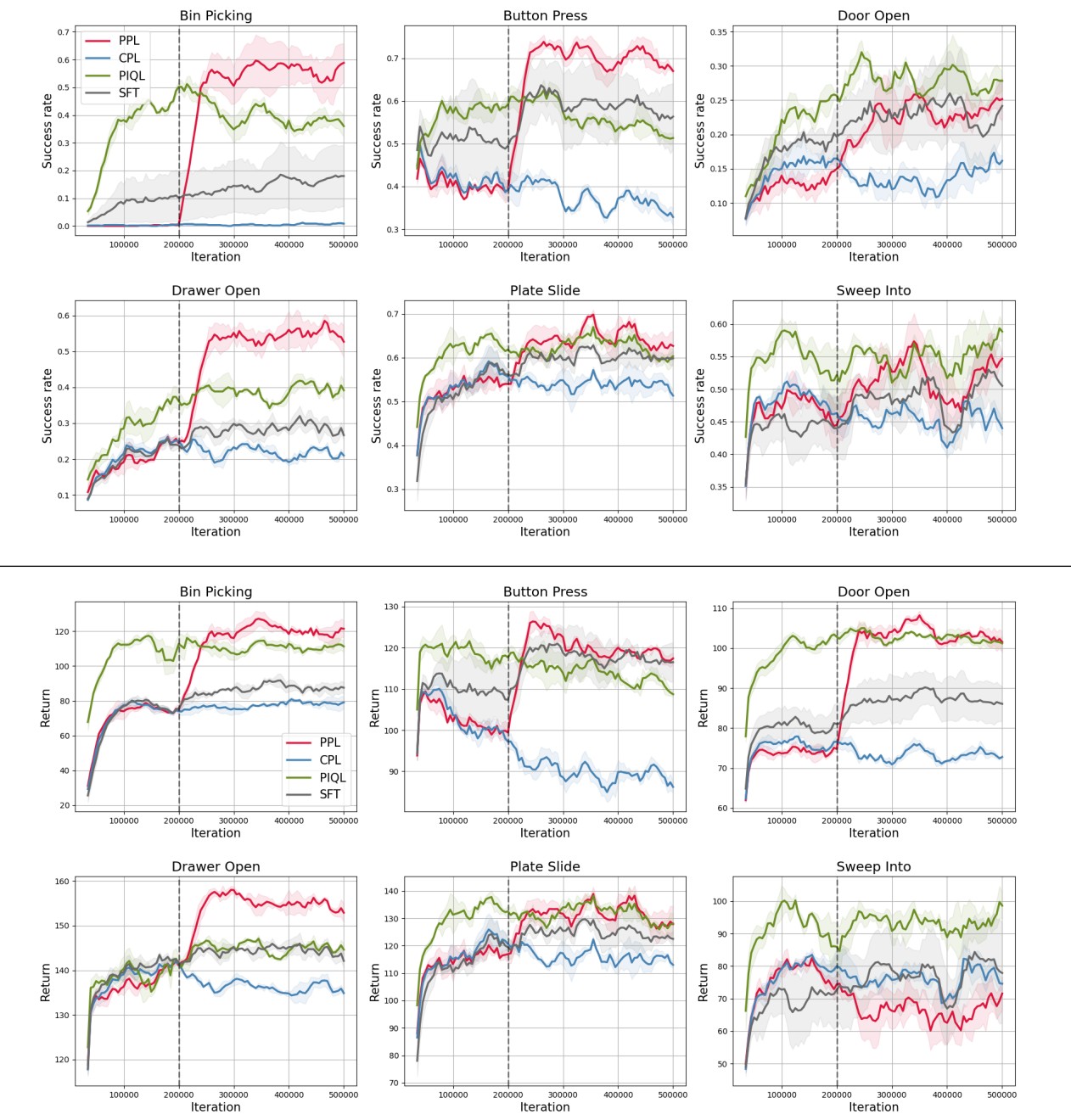

Figure 13: Performance comparison of different methods on the `Heterogeneous Dense` dataset across six MetaWorld tasks.

## F.4. `Heterogeneous Sparse` Offline Dataset

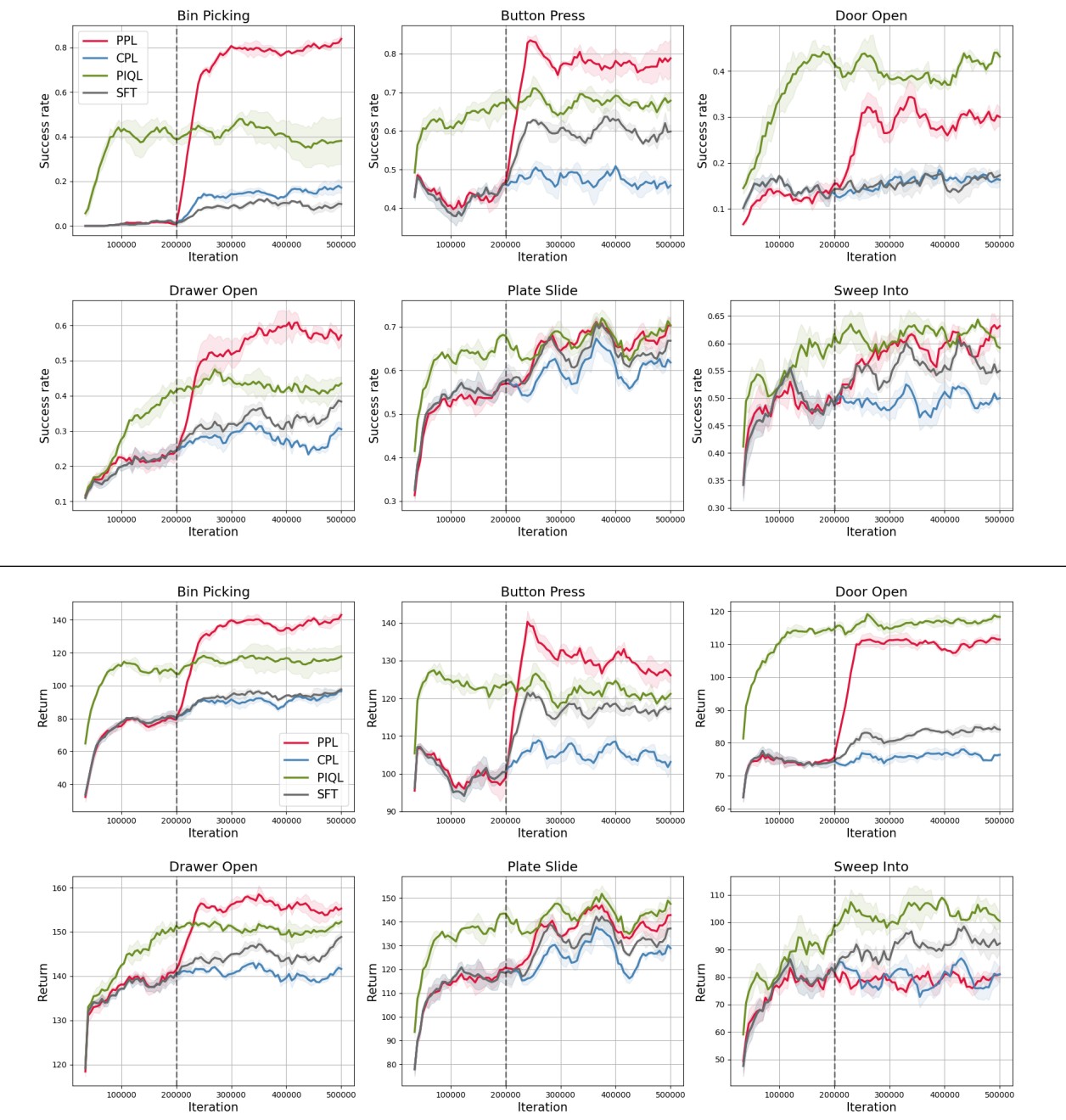

Figure 14: Performance comparison of different methods on the `Heterogeneous Sparse` dataset across six MetaWorld tasks.

# G. Comparison with Deterministic Pseudo-labels (Section 4.3)

## G.1. `Homogeneous Dense` Offline Dataset

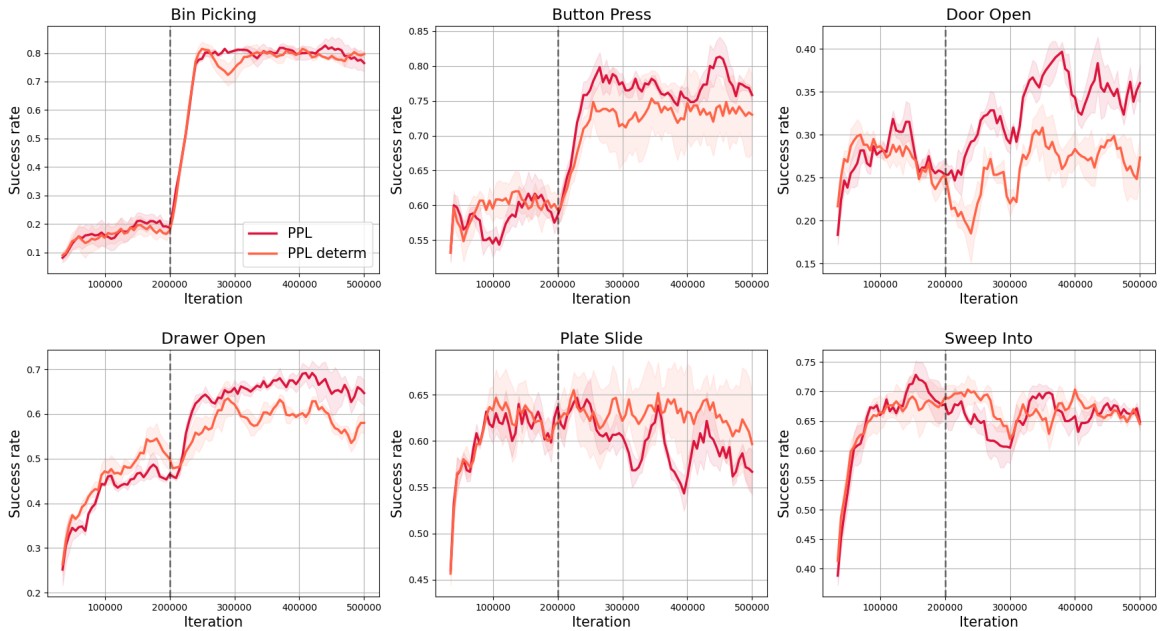

Figure 15: Comparison of PPL and PPL-deterministic on the `Homogeneous Dense` Offline Dataset.

## G.2. `Heterogeneous Dense` Offline Dataset

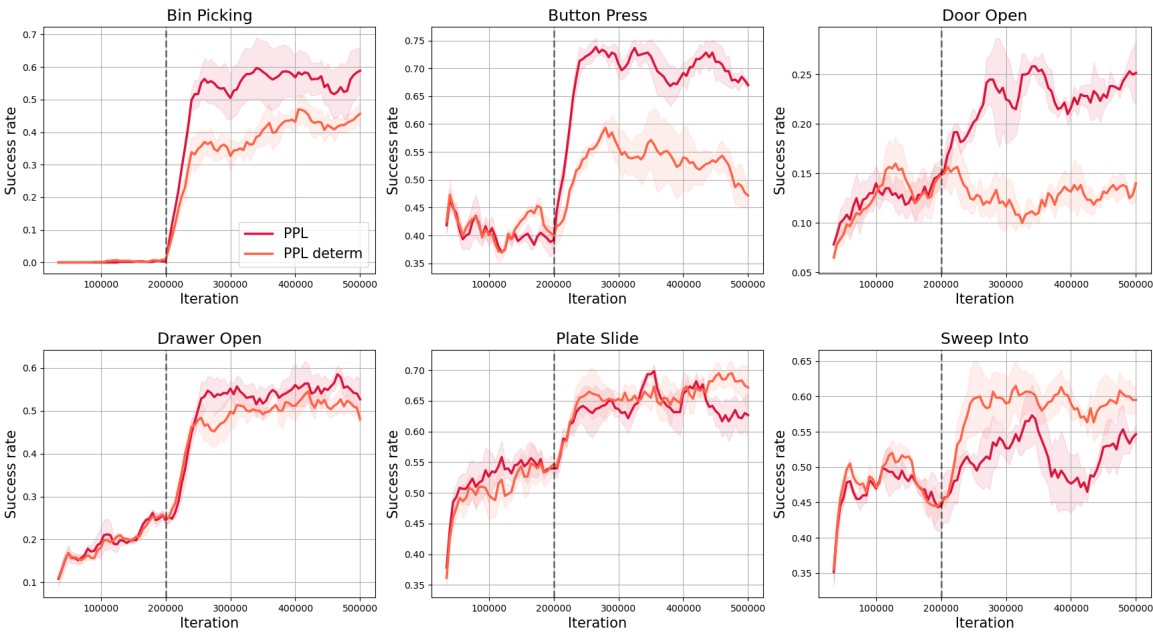

Figure 16: Comparison of PPL and PPL-deterministic on the `Heterogeneous Dense` Offline Dataset.

# H. Experimental Results on Online Implementation (Section 4.4)

## H.1. Online Learning Curves

We evaluated the performance of PPL in an online setting across five MetaWorld tasks. The number of preference queries (#Pref) varied for each environment based on the quantities used in PEBBLE, and these differences are illustrated in each plot.

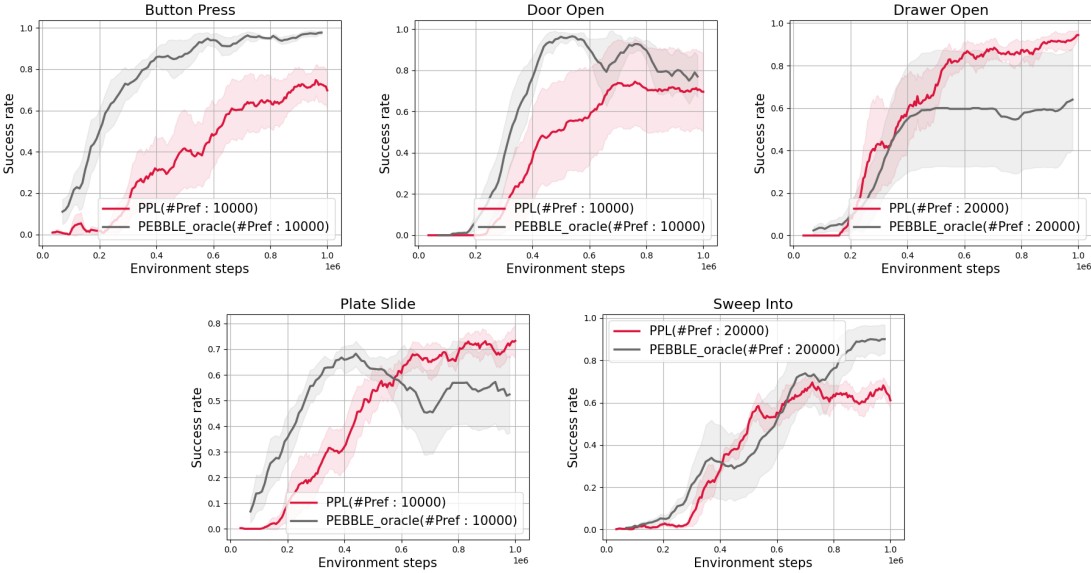

Figure 17: PPL and PEBBLE learning curves in online learning.

## H.2. Ablation on Preference Query Count

We evaluate the performance of PPL over iterations with different numbers of preference queries (#Pref). Overall, increasing the number of preference queries leads to improved performance, demonstrating the benefit of richer preference feedback in online learning.

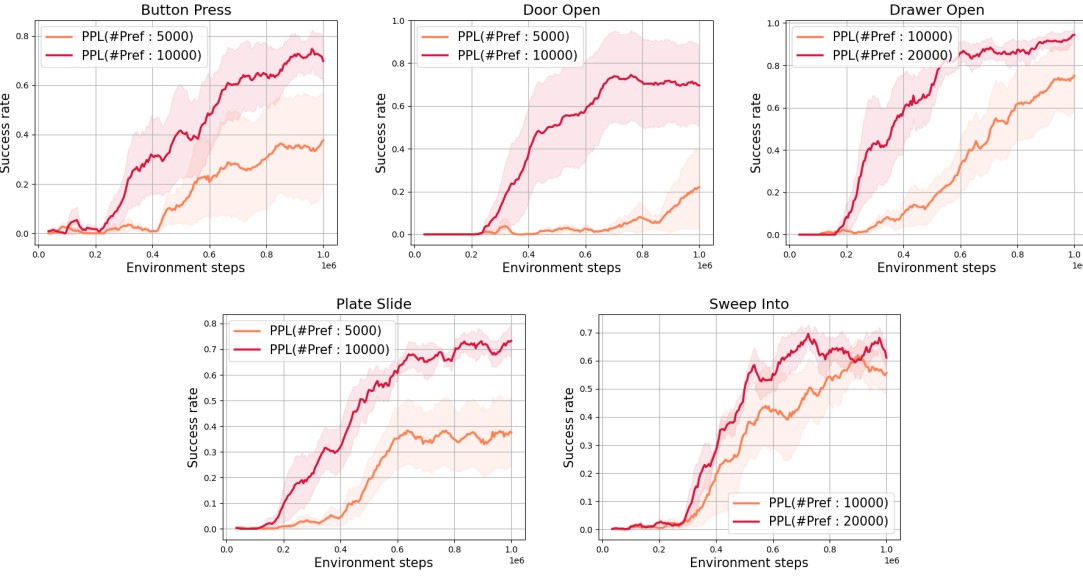

Figure 18: Effect of preference query count in online learning.

## H.3. Ablation on Rollout Length

We analyze the impact of different rollout lengths $L$ on the performance of PPL in an online RLHF setting across five MetaWorld tasks. Each plot compares the success rate over training iterations for three rollout lengths: $L = \{5, 10, 20\}$.

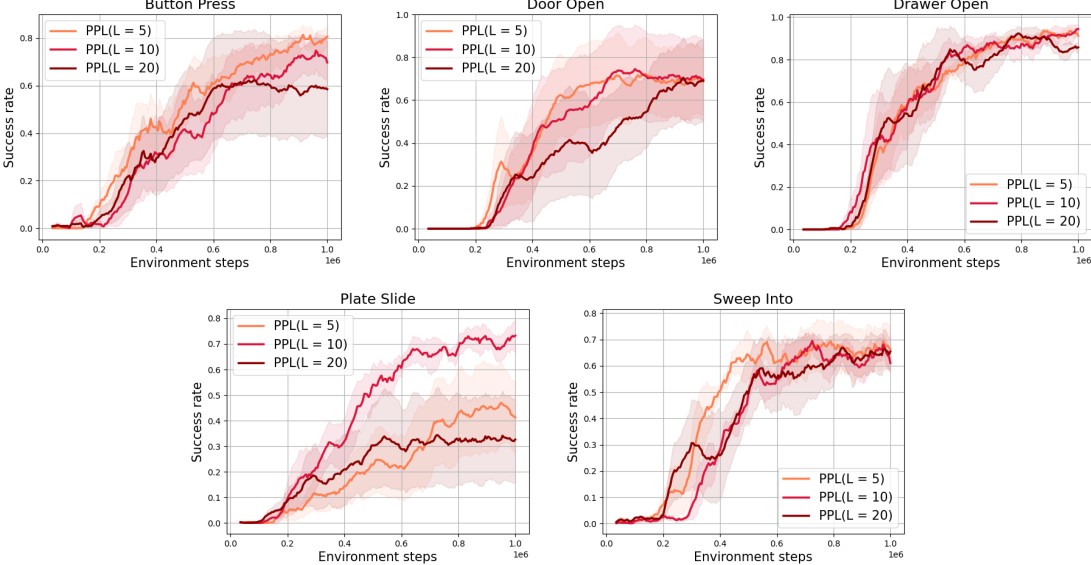

Figure 19: Effect of rollout length in online learning.

