# OpenReview forum: "Policy-labeled Preference Learning: Is Preference Enough for RLHF?"
_ICML.cc/2025/Conference — ICML 2025 spotlightposter_

### Official Review · Reviewer_4oay · 2025-03-09

**Overall Recommendation:** 3

**Summary:**

This paper introduces PPL, a new method for learning policies from human preferences based on the regret based model of preferences. The authors note a key distinction between the regret model used in prior work (CPL by hejna et al.) and propose an improvement upon it to consider the current policy. The authors evaluate the method on a number of tasks with heterogeneous and homogenous policy data.

**Claims And Evidence:**

The claims of the paper are generally well supported. The authors do a good job:
* Explaining the differences with prior work
* Deriving a new update rule
* evaluation on a nubmer of tasks.

There are few missing citations that I believe should be added, and adjustments to be made which I left to the essential references section.

**Essential References Not Discussed:**

**Regret-based Preference Model**

The regret based preference model was not first introduce by Hejna et al. It was instead introduce by Knox & Hatgis-Kessell et al. in "Models of human preference for learning reward functions" published in 2023. This was further explored in "Learning optimal advantage from preferences and mistaking it for reward" by Knox et al. It would be nice if the authors could attribute the regret-based preference model to these works, rather than CPL which used it to derive their algorithm.

**Theoretical Justification**

* The derivation of Lemma 3.2 bears high similarity to "From  to : Your Language Model is Secretly a Q-Function" by Rafailov & Hejna et al. published in COLM 2024, and should probably be attributed there.

* Theorem 3.4 seems highly related to Proposition A.1. from "Show, Don't Tell: Aligning Language Models with Demonstrated Feedback" from Shaikh et al., published in ICLR 2024. The only distinction between this and Theorem 3.4 is the use of the full discounted MDP. However, up to notation and changing to the discounted KL, they are the same.

**Experimental Designs Or Analyses:**

**Reproduction**

The experimental design is good -- it considers two cases of preference data from heterogeneous or homogeneous policies. The authors went the extra mile reproducing the original results from CPL.

However, the performance on the authors own datasets for CPL is far worse than on the original CPL datasets, in fact so much so that SFT often performed better than CPL. In particular, performance doesn't go up at all for CPL in Fig 5 even in the homogenous case. One question I have is if the authors re-trained the oracle policy with the behavior segments used for data added to the replay buffer as done in the original CPL paper to ensure more accurate advantage function estimates.


**Performance**
The authors method performs favorably in comparison to prior methods.

**Methods And Evaluation Criteria:**

The authors present a nice small-scale example of where the previous preference model might be incorrect in Fig 3.

The method's derivation is intuitive and makes sense.

One weakness I would like to point out is that though the authors spend time deriving their method, the final practical implementation is not easily grasp-able from the main body of the paper, and is instead most easily found in the appendix. I would encourage the authors to include a derivation of the final objective within the text of the main paper, I think this would make the work much more understandable and stronger.

**Other Comments Or Suggestions:**

* Include a discussion of why the presented theory is valueble. If space is a limitation, I would rather have this in the main text and delegate some of the intermediary results to the appendix.

* Include the final practical objective in the main text. I believe this is currently in a later section of the appendix.

**Other Strengths And Weaknesses:**

Strengths: The paper identifies a theoretical weakness, and develops a method to address it and demonstrates improved performance.

Weaknesses: some questions remain around the dataset generation and evaluation procedure. The presentation could be improved, particulary in providing the final practical method and explaining the theory. Finally, some critical citations are missing.

Should the authors address the weaknesses listed (which I believe should be very do-able, I am willing to raise my score.)

**Questions For Authors:**

* Why is CPL performance on the re-created datasets so low if the authors used the same procedure?
* Could the authors clarify if they trained their label-generating oracle policy with the trajectories used to sample preferences in the replay buffer of the SAC policy? This would lower the TD-error when estimating the advantage.
* Why are only positive segments used for Theorem 3.6?

**Relation To Broader Scientific Literature:**

The authors sufficiently cover related work, aside from the points brough up in the next section of the review.

**Theoretical Claims:**

* The theoretical claims around preference models are supported by apt examples. Note that I did not carefully check all proofs provided in Section 3.2, but I did scan the derivations in the Appendix. I believe some of the results need proper attribution to prior works (see later section).

* While the theoretical results are interesting, it was less clear form an intuitive perspective why we should care about the theoretical results. If the authors could provide more support for what insights these are.

* Some of the results could be explained a bit more. Why is the expectation in Theorem 3.6 taken over only preferred segments? This does not seem to be explained.

* I do not believe that all claims made desrve the level of distinction provided. For example, I'm not sure that Theorem 3.4 constitutes as theorem. A similar result was considered a Proposition in "Show, Don’t Tell: Aligning Language Models with Demonstrated Feedback" by Shaikh et al.

---

> ### Author Rebuttal · Authors · 2025-04-01
>
> We sincerely thank the reviewers for their time and thorough evaluation of our paper. We have organized our responses to your comments below. If any of our responses fail to address the intent of your questions or if you have remaining concerns, please let us know.
>
> **1. Final practical implementation**
>
> We agree that it would be beneficial for readers to have a clearer view of the final implementation in the main text, and we will revise to address this.
>
> **2. Regret-based preference model attribution**
>
> Thank you for clearly outlining the relationships with the literature. We will revise and add a reference to [Knox et al. 2022] to clearly attribute the origin of the regret-based preference model as referenced in CPL.
>
> **3. Including the meaning of theoretical results**
>
> We appreciate your insight that providing intuitive explanations for our theoretical results can further enhance the reader's understanding. We have endeavored to include interpretive statements following each theoretical result. Could you please kindly indicate if there are specific parts of the theoretical results that you believe would benefit from additional clarification?
>
> **4. Theoretical justification**
>
> - **(Lemma 3.2):**
> We have confirmed that [Rafailov et al. 2024a] derived the same theoretical results as [Rafailov et al. 2024b], and we will cite this in the paragraph explaining Lemma 3.2.
>
> - **(Theorem 3.4):**
> While [Shaikh et al. 2024] presented their results in a contextual bandit setting, our Theorem 3.4 extends this to MDPs. This extension is non-trivial due to the presence of stochastic transitions. Though [Zeng et al 2024] achieved results more similar to ours than [Shaikh et al. 2024], their proof applies only to deterministic transitions. Thus, Theorem 3.4 makes a distinct theoretical contribution by addressing environmental stochasticity.
>
> **5. Theorem 3.6**
>
> First, Corollary 1 of [Hejna et al. 2024], which motivated Theorem 3.6, showed that maximizing the optimal advantage is equivalent to learning the same optimal policy as maximizing the reward. In this context, Theorem 3.6 was written to make it intuitively easier to understand what kind of policy is learned from a reinforcement learning perspective when maximizing negative regret. The reason for considering only the preferred segment here is to interpret this intuitive meaning more clearly. For less preferred segments, the PPL objective can be expressed as:
>
> $\arg \min\_{\pi\_{\psi}}\Big(\mathbb{E}\_{\zeta^- \sim \mathcal{D}}[-Reg^{\pi^-}\_{\pi\_{\psi}}(s^- ,a^-)-\alpha \log \pi\_{\psi}(a^- |s^-)]\Big) \equiv \arg\max\_{\pi\_{\psi}}\Big( \mathbb{E}\_{\zeta^- \sim \mathcal{D}}[\bar{D}\_{KL}(\pi^- || \pi\_{\psi} ; s^-, a^-)]\Big)$
>
> However, since the left-hand side represents cost minimization rather than reward maximization in traditional reinforcement learning, it is difficult to interpret intuitively. To avoid this ambiguity, we only dealt with the preferred segment perspective.
>
> **6. Dataset Generation and Evaluation Procedure**
>
> - **Dataset Generation:** During SAC training, we saved checkpoints of models showing 20%, 50%, and 100% success rate. We loaded policy model to generate 20K segments, and for dense cases, randomly sampled 2.5K segments. For labeling procedure, we used $V^{\pi^*}(s_k)-V^{\pi^*}(s_0) + \sum_{t=0}^{k-1} r(s_t,a_t)$ from CPL, and calculated $V^{\pi^*}$ using 64 MCMC samples with a 100% success rate critic $Q^{\pi^*}$. For fair comparison, all algorithms were trained with the same labels, and when policy labels were needed in PPL, logits were calculated using policy information stored during rollout.
>
> - **Evaluation Procedure:** We conducted evaluations following the same method as CPL: (1) run 25 evaluation episodes, (2) calculate averages across 8 neighboring checkpoints, (3) average across 4 seeds, and (4) report the maximum values in the table.
>
> For clarity of experimental design, we will add the provided response to the appendix.
>
> **7.Retrain the SAC oracle for accurate advantage estimation**
>
> In the CPL dataset creation process, data segments are added to the replay buffer of the oracle SAC and then re-trained, which helps lower the TD error and improve the accuracy of advantage estimation. While we do not adopt the same procedure, we used only a single SAC oracle and trained all algorithms with the same labels.
> Therefore, it is difficult to conclude that the inaccuracy of advantage estimates caused performance degradation only for CPL.
>
> One likely reason for CPL's struggle with noisy labels appears to be similar to the likelihood mismatch issue discussed earlier. Since CPL does not consider policy labels, it assumes all training data segments originate from an optimal policy. We suspect that when noisy labels occur, suboptimal data segments might be incorrectly treated as if they were generated by the optimal policy, potentially making the model more sensitive to noise and degrading its learning performance.

---

> > ### Comment · Reviewer_4oay · 2025-04-03
> >
> > I would like to thank the authors for their responses and work in rebuttal in addition to answering my questions!
> >
> > It still seems like the authors do not follow the exact procedure used in CPL for generating the data (namely they do not minimize the bellman error of the Q function used to estimate A on the rollout datasets). At a minimum, it would be good to state this explicitly as the draft currently states that the same data generation procedure was used "we follow CPL’s preference dataset generation procedure" in line 358. At best, it would be good to know if this could explain CPLs low performance versus other explanations.

---

> > > ### Author Response · Authors · 2025-04-03
> > >
> > > We sincerely thank the reviewer for the prompt and thoughtful feedback. To better reflect the differences you pointed out, we will revise the manuscript as follows:
> > >
> > > **Line 356–359 (Left column):**
> > >
> > > To evaluate performance on offline datasets generated from diverse policies, we aimed to follow CPL’s preference dataset generation procedure. However, there are two key differences in our implementation of the critic. First, we utilize raw rollout data without any trajectory truncation. Second, whereas CPL applies a specific technique to reduce TD-error by re-training the critic with all rollout data added to the replay buffer, we generated preference labels *without such retraining*. As a result, our labels may be noisier than those in CPL. Nevertheless, to ensure a fair comparison, all algorithms were trained using the same set of labels. For further details, please see Appendix E.5.
> > >
> > > **Line 386 (Left column):**
> > >
> > > We suspect that CPL’s lower performance on our dataset may partially result from the absence of this retraining technique to reduce TD-error. However, since all algorithms were trained using the same labels, we believe this performance gap is better explained by CPL’s sensitivity to label noise. This sensitivity likely stems from CPL’s implicit assumption that all training trajectories are generated by an optimal policy.

---

### Official Review · Reviewer_xoy5 · 2025-03-13

**Overall Recommendation:** 4

**Summary:**

The paper proposes Policy-labeled Preference Learning (PPL) to mitigate what it calls “likelihood mismatch” in RLHF. The authors illustrate how policy labels (i.e., knowledge of which behavior policy generated which trajectory) can help disentangle suboptimal policy actions from stochasticity in the environment and thereby improve preference modeling. They further introduce contrastive KL regularization as a tool to align the learned policy with the policies associated with higher-preference trajectories.


**Update after Rebuttal:** I'm keeping my positive score.

**Claims And Evidence:**

In my opinion, the main claims of the submission are well-supported.

**Essential References Not Discussed:**

As far as I am aware related work has been appropriately discussed.

**Experimental Designs Or Analyses:**

I read through the experimental details and superficially skimmed through the additional details in the appendix. Question to the authors: Could you please clarify whether in the experiments of Section 4.2 you assume access to the policy labels? This is not fully clear to me from the text.

**Methods And Evaluation Criteria:**

The authors perform experiments on six MetaWorld environments and test both homogenous and heterogenous offline dataset (homogenous/heterogenous here refers to the generating behavior policies). MetaWorld is a robotics benchmark, which is a reasonable choice in the context of RLHF. However, the authors begin the paper with a discussion around LLM fine-tuning so that there is a slight disconnect between how the paper positions itself in the introduction and the type of experiments it conducts.

**Other Comments Or Suggestions:**

In my opinion, "Is Preference Enough for RLHF?" is a slightly misleading title, as you are explicitly working under the premise that examples are generated using a heterogenous/diverse policies. You are not discussing the limitations of RLHF w.r.t. pairwise preferences more broadly. This part of the title could maybe be made less broad and more accurate.

**Other Strengths And Weaknesses:**

### Strengths
1. The authors highlight the issue of likelihood mismatch due to heterogenous behavior policies.
2. Using the regret w.r.t. the behavior policy in contrats to the optimal advantage as in CPL appears like a natural approach to address the challenge of diverse generating policies.
3. The experiments support the claims that incorporating policy labels can improve substantially in certain cases.

### Weaknesses
1. One key assumption is that either we know the precise policy that generated each trajectory or we can assign a "pseudo-label". In many real-world applications (especially large, unstructured offline data), extracting reliable policy labels is nontrivial. The paper briefly uses deterministic pseudo-labels but does not deeply analyze the accuracy needed or possible mismatch when pseudo-labels are very noisy as far as I am aware.
2. A minor weakness is that that the role of coverage is not sufficiently discussed in the paper in my opinion. Likelihood mismatch seems to only be an issue when coverage is incomplete (or biased). Discussing this in more detail especially seems important around Figure 3.

**Questions For Authors:**

The experiments focus on comparing the performance of PPL only with CPL, P-IQL and SFT. For the case of heterogenous data, PPL is only compared to CPL in the main text and additional plots shown in the appendix. Is there a reason why you are not comparing with DPO and DPPO? Are there other offline RL methods that explicitly address similar issue to the likelihood mismatch (i.e., policy mismatch) that you could compare to?

**Relation To Broader Scientific Literature:**

The paper can be viewed as a refinement of contrastive preference learning (CPL) by explicitly taking into account the behavior policy and the "likelihood mismatch" that can arise.

**Theoretical Claims:**

I read through the proofs of Lemma 3.2, 3.3 and Theorem 3.4 and found no issues.

---

> ### Author Rebuttal · Authors · 2025-03-31
>
> We sincerely thank the reviewers for their time and thorough evaluation of our paper. We have organized our responses to your comments below. If any of our responses fail to address the intent of your questions or if you have remaining concerns, please let us know.
>
> **1. Paper position disconnection with introduction**
>
> To emphasize that the PPL algorithm learns policies without explicitly learning a reward function, we mentioned in the Introduction that we are adopting the Direct Preference Optimization framework. Although the DPO paper, which introduced this framework, has been studied in the LLM domain, we noted in line 25 (right column) that its deterministic transitions make it an unsuitable setting for demonstrating the advantages of PPL.
>
> **2. Label usage in experiments**
>
> In the experiments in Section 4.2, PPL assumed that policy labels are accessible. However, since extracting reliable policy labels is nontrivial, in Section 4.3 we designed and experimented with PPL-deterministic, which utilizes pseudo labels.
>
> **3. Analysis under noisy label**
>
> Thank you for your comment regarding noisy pseudo-labels. We are currently conducting experiments in which we add Gaussian noise, $\mathcal{N}(0, \sigma^2)$to the ground-truth labels. In the button-press environment, we observed that performance degrades as the noise level $\sigma$ increases. Specifically, at $\sigma$=0.1, the performance was lower than that of PPL but higher than that of PPL-deterministic. However, at $\sigma$=0.3, the performance dropped below that of PPL-deterministic. This indicates that if the pseudo-label is not sufficiently close to the policy label, its performance will be inferior to the proposed PPL-deterministic, thereby confirming that the pseudo-label used in PPL-deterministic is a reasonable alternative.
>
> **4. Likelihood mismatch and role of coverage**
>
> As described in Figure 3 and in the section on likelihood misalignment, the likelihood mismatch issue does not primarily arise from dataset coverage. Instead, it occurs when the algorithm assumes that trajectories are always generated by the optimal policy. As shown in Figure 3, let us consider the black trajectory is generated by the optimal policy, while the red trajectory is produced by a suboptimal policy. Let's consider the case where all four trajectories—which ensure complete data coverage—are included in the dataset. Without the policy label, the estimated MDP appears as shown in the right panel of Figure 3, and it is evident that it still misinterprets the ground-truth MDP depicted in the left panel.
>
> **5. Misleading title**
>
> Thank you for your detailed feedback. However, we believe that the scenario considered in our work—where data is generated by heterogeneous/diverse policies—is more justified and encompasses a broader range of realistic situations than merely assuming a homogeneous dataset. Previous studies using offline datasets have not taken into account the policy that generated the dataset. In contrast, our paper highlights this point and proposes an algorithm that leverages policy information, thereby emphasizing that in RLHF settings, both preference and policy information should be considered. We consider the current title appropriate, but if you feel that this context is not sufficiently conveyed, we would greatly appreciate any further suggestions.
>
> **6. Comparing with DPO and DPPO**
>
> Since the [Rafailov et al 2024] presents an algorithm designed for bandit settings, it could not be directly applied to sequential tasks such as robotic manipulation, and therefore was not compared separately. In the case of the DPPO, since CPL assumes a fixed standard deviation Gaussian for the policy, its loss simplifies to an L2 loss between actions—identical to the implementation used in DPPO [An et al. 2023]. We confirmed this similarity based on the official CPL GitHub repository, so no separate comparison was performed.
>
> **7. Other offline likelihood (policy mismatch) literature**
>
> To the best of our knowledge, we have not found any offline RL algorithm that considers likelihood mismatch or policy mismatch.
>
> **References**
> * [Rafailov et al 2024] : Rafailov, R., Sharma, A., Mitchell, E., Manning, C. D., Ermon, S., and Finn, C. Direct preference optimization:Your language model is secretly a reward model. Advances in Neural Information Processing Systems, 36.
> * [An et al 2023] : An, Gaon, et al. "Direct preference-based policy optimization without reward modeling." Advances in Neural Information Processing Systems 36 (2023): 70247-70266.

---

### Official Review · Reviewer_MyYU · 2025-03-14

**Overall Recommendation:** 4

**Summary:**

The paper proposes a novel approach to preference-based RL, grounded in regret-based modeling of human preferences. Unlike prior work that also uses regret-based modeling, the paper explicitly labels the behavior policy. This, as argued by the authors, is important in order to  resolve the likelihood mismatch issues when utilizing regret-based modeling of human preferences. The paper provides a formal analysis of their novel framework, which yields a new algorithmic approach to preference-based RL. Finally, extensive experiments are conducted to showcase the benefits of the approach.

---
## update after rebuttal

I thank the authors for their response. I will keep my original score.

**Claims And Evidence:**

In my opinion, he set of results in this paper is quite convincing and it clearly demonstrates the utility of the proposed approach. The paper provides an in-depths analysis (both theoretical and empirical) of the considered problem setting. I also appreciate that the intuitive examples in Section 3 which motivate the approach.

**Essential References Not Discussed:**

The background and the main reference that this work extends is explained well. However, it would be great if the authors could include a separate related work section and provide an overview of the literature on RLHF and DPO. I can see several important references cited in the related work, including those that discuss MDP extensions of DPO (e.g., Rafailov et al. 2024a, Nika et al. 2024, etc.). However, these don't seem to be extensively discussed in the main paper.

**Experimental Designs Or Analyses:**

The experimental test-bed is rigorous and follows closely related prior work on preference-based RL. The experiments are extensive and test various aspect of the proposed approach. I believe that the experiments are well-designed and I didn't spot major issues with the experimental analysis.

**Methods And Evaluation Criteria:**

The paper provides both a theoretical investigation of the problem setting and an empirical evaluation of the proposed approach. The test-bed is primarily based on simulation-based robotics tasks. It would be interesting to see whether the results generalize to other domains, but I don't think this is necessary given that the contributions are substantial.

**Other Comments Or Suggestions:**

Suggestions:
- I didn't initially understand how Eq. (3) was obtained from (2). I noticed the proof in B.1, but it seems that $Q_*^{\pi}$ is only introduced afterwards. Might be good to explain what  $Q_*^{\pi}$ denotes beforehand.
- Minor typo in Lemma 3.3: 'Let $\pi^*$ is' -> 'Let $\pi^*$ be'

**Other Strengths And Weaknesses:**

The paper is well-written, easy to follow and enjoyable to read. The contributions are non-trivial in my opinion; while the main idea behind the proposed approach is quite intuitive, to my knowledge it is conceptually novel. The theoretical analysis is also novel, although the proof techniques seem to be based on standard arguments. The experimental analysis is also thorough. Overall, I found the results quite interesting and convincing.

**Questions For Authors:**

Please see my comments above. I don't have specific clarification questions at the moment.

**Relation To Broader Scientific Literature:**

This work is broadly related to the literature on AI alignment, focusing more specifically on preference-based RL. The proposed approach could potentially be more broadly applicable, e.g., for training LLMs in multi-turn dialogues. However, the current results primarily focus on RL and manipulation policies in robotics.

**Theoretical Claims:**

I've taken a look at the theoretical claims, including the proofs in the appendix. The claims and the proofs seem to be correct.

---

> ### Author Rebuttal · Authors · 2025-03-31
>
> We sincerely thank the reviewers for their time and thorough evaluation of our paper. We have organized our responses to your comments below. If any of our responses fail to address the intent of your questions or if you have remaining concerns, please let us know.
>
> **1. Generalization to other domains**
>
> Since PPL is particularly beneficial for sequential and stochastic scenarios, we determined that a robotic manipulation task is more appropriate than a language domain with deterministic transitions, and thus conducted our experiments accordingly. If extended to the language domain, exploring the multi-turn LLM scenario you suggested would be an excellent direction for future work.
>
> **2.LLM literature reference**
>
> Recent research on RLHF has been particularly active in the LLM domain, and our PPL algorithm which directly learns policies without explicitly learning a reward function shares similarities with the Direct Preference Optimization framework. We briefly mention this connection in the main text. However, since our experiments focus on sequential RL tasks such as robotic manipulation, we decided not to deeply discuss related work specific to LLM tasks to avoid reader confusion.
>
> **3. Response to [Other Comments and Suggestions]**
>
> In Equation (2), we noticed that the derivation of Equation (3) is referenced as being in Appendix B.1, but that reference appears about one paragraph after the equations are presented. To reduce any potential confusion for readers, we will adjust the sentence placement.
> Additionally, the explanation for the subscript * is provided later than its initial usage. Since removing the subscript * does not change the original meaning of Equation (2), we will remove it from that equation. We will also correct the typo in Lemma 3.3 as you pointed out.

---

### Official Review · Reviewer_Gpnc · 2025-03-15

**Overall Recommendation:** 4

**Summary:**

This paper proposes an algorithm called policy-labeled preference learning where the preferences are assumed to be formed by a regret-based metric and the preference data are labeled by the policy that generated the trajectories. The paper did a very good job explaining why the regret-based preference model may be preferable and how it is different from the advantage-based model in the literature. The results are very strong against the state-of-the-art baselines.

**Claims And Evidence:**

The claims made in the submission are supported by clear and convincing evidence. Overall, I find the paper to be difficult to argue against. Kudos to the authors.

**Essential References Not Discussed:**

N/A

(though, just like many other recent RLHF papers, this one also mostly neglects the pre-InstructGPT preference-based reward learning literature other than the Christiano et al. 2017 paper)

**Experimental Designs Or Analyses:**

The experimental designs and analyses make sense.

While there are a few cases where baselines perform better than PPL, in most cases PPL outperforms the others. Such consistent improvement is not very common in the RLHF literature, so I would say the results are very strong.

**Methods And Evaluation Criteria:**

The methods and the evaluation criteria make sense. While reading the paper, I always had in mind that the proposed method will not work for the cases where there already exists an offline dataset without policy labels. However, the paper considered this cases too and proposed a heuristic method to get around that issue. It also conducted experiments under that setting.

**Other Comments Or Suggestions:**

There are a few typo-like issues:

1) In the equation before Section 3.2, the term inside the summation must be inside a parenthesis as otherwise the second term is outside of the summation.
2) In line 313 (right column), there seems to be an extra word that should be deleted.
3) In line 379 (left column), "implementation" is misspelled.

Finally, it might be a good idea to divide Figure 4 into three separate figures so that the histograms are separately visible -- currently, their overlapping regions are difficult to distinguish, if not impossible.

**Other Strengths And Weaknesses:**

N/A

**Questions For Authors:**

N/A

**Relation To Broader Scientific Literature:**

The paper builds upon the CPL algorithm. One thing that I am not sure about is: the paper cites "Learning Optimal Advantage from Preferences and Mistaking It for Reward" by Knox et al. which also considers a regret-based method. Does their method also suffer from the same problem as the CPL paper, i.e., do they also present the regret-based model as equivalent to the advantage-based model? If so, perhaps that should also be mentioned when this limitation of CPL is discussed in the paper. If this is not the case, a clarification of why the regret-based model of this paper is different/better is needed, as otherwise the novelty becomes thinner (though the method and the theoretical analyses are still novel and I would still argue for the acceptance of the paper).

**Theoretical Claims:**

I did not check the proofs as they are in the Appendix, but the claims intuitively made sense to me.

---

> ### Author Rebuttal · Authors · 2025-03-31
>
> We sincerely thank you for your time and effort in reviewing our paper. We have organized our responses to your comments below. If any of our reconstructed responses miss the intent of your questions or if there are remaining concerns, please let us know so we can address them.
>
> **1. Difference between PPL and cited literature**
>
> The regret-based preference model used in the CPL algorithm, which defines regret in terms of the optimal advantage function, was first introduced in [Knox et al. 2022]. Similarly, [Knox et al. 2024] interprets preferences using the same optimal advantage-based regret model, arguing that the Bradley-Terry model yields an approximation of the optimal advantage function rather than a true reward function. The regret preference models used in CPL, [Knox et al. 2022] and [Knox et al. 2024] are both essentially based on optimal advantage, which is a different design from the regret model proposed in PPL.
>
> **2. Response to [Other Comments and Suggestions]**
>
> Thank you for pointing out the typo, and I will incorporate it into the revised version for better clarity.
> Regarding Figure 4, it was designed as a single plot to enable an at-a-glance comparison of the distributions across the three datasets. However, we recognize that overlapping colors make it difficult to distinguish the overlapping regions. Given the space constraints in the main text, we will either separate the plots or incorporate an envelope to improve boundary delineation.
>
> **References**
> * [Knox et al 2022] : Knox, W. Bradley, et al. "Models of human preference for learning reward functions." arXiv preprint arXiv:2206.02231 (2022).
> * [Knox et al 2024] : Knox, W. Bradley, et al. "Learning optimal advantage from preferences and mistaking it for reward." Proceedings of the AAAI Conference on Artificial Intelligence. Vol. 38. No. 9. 2024.

---

### Decision · Program_Chairs · 2025-05-01

**Decision:**

Accept (spotlight poster)

**Comment:**

The paper proposes an interesting regret based metric for modeling human preferences. All the reviewers agreed that the theoretical work is sound and the experimental results highlight the importance of the work. The proposed model is a viable alternative to existing advantage-based preference model, and can also be useful beyond deep RL settings.